# HIERARCHY-OF-GROUPS POLICY OPTIMIZATION FOR LONG-HORIZON AGENTIC TASKS

**Shuo He[1], Lang Feng[1], Qi Wei[1], Xin Cheng[1], Lei Feng[2],\*Bo An[1]**
Nanyang Technological University[1], Southeast University[2]
shuohe123@gmail.com, fenglei@seu.edu.cn

## ABSTRACT

Group-based reinforcement learning (RL), such as GRPO, has advanced the capabilities of large language models on long-horizon agentic tasks. To enable more fine-grained policy updates, recent research has increasingly shifted toward *stepwise* group-based policy optimization, which treats each step in a rollout trajectory independently while using a memory module to retain historical context. However, we find a key issue in estimating stepwise relative advantages, namely *context inconsistency*, where steps within the same group may differ in their historical contexts. Empirically, we reveal that this issue can lead to severely biased advantage estimation, thereby degrading policy optimization significantly. To address the issue, in this paper, we propose Hierarchy-of-Groups Policy Optimization (HGPO) for long-horizon agentic tasks. Specifically, within a group of rollout trajectories, HGPO assigns each step to multiple hierarchical groups according to the *consistency* of historical contexts. Then, for each step, HGPO computes distinct advantages within each group and aggregates them with an adaptive weighting scheme. In this way, HGPO can achieve a favorable bias-variance trade-off in stepwise advantage estimation, without extra models or rollouts. Evaluations on two challenging agentic tasks, ALFWorld and WebShop with Qwen2.5-1.5B-Instruct and Qwen2.5-7B-Instruct, show that HGPO significantly outperforms existing agentic RL methods under the same computational constraints. Code is available at `https://github.com/langfengQ/verl-agent/tree/master/recipe/hgpo`.

## 1 INTRODUCTION

Versatile agents powered by Large Language Models (LLMs) can perceive, reason, and act in complex, open-ended environments (Achiam et al., 2023; Team et al., 2023; Yang et al., 2024; Liu et al., 2024). Representative applications include embodied assistants navigating simulated homes (Shridhar et al., 2021; Li et al., 2024), web navigators completing browsing tasks (Furuta et al., 2024; Zheng et al., 2024; Gou et al., 2025), and autonomous explorers in interactive computer games (Wang et al., 2024a;b). Beyond language and vision understanding, such agents are expected to perform long-horizon planning and robust decision-making.

Deep reinforcement learning (RL) (Sutton & Barto, 2018) has emerged as a key paradigm for enhancing agent performance in the post-training stage (OpenAI, 2024; Guo et al., 2025). In particular, group-based RL methods such as RLOO (Kool et al., 2019; Ahmadian et al., 2024), GRPO (Shao et al., 2024), DAPO (Yu et al., 2025c), Clip-Cov (Cui et al., 2025), and GSPO (Zheng et al., 2025) have demonstrated strong performance in large-scale RL training while requiring fewer computational resources. These methods have proven effective in single-turn tasks such as mathematical reasoning (Liu et al., 2025; Yu et al., 2025c) and code generation (Wei et al., 2025a). To extend this paradigm to multi-turn settings, approaches such as RAGEN (Wang et al., 2025d) and Search-R1 (Jin et al., 2025a) adopt a *trajectory-wise* policy optimization framework, which concatenates environment states and model outputs across turns to enable multi-turn rollouts. However, this framework suffers from a major limitation: the effective context length grows rapidly with the number of interaction turns, leading to severe context explosion.

---

*Corresponding author

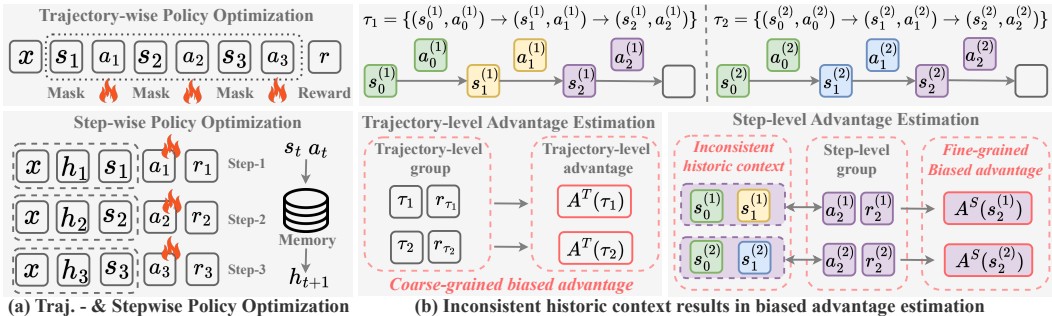

Figure 1: Figure (a) compares trajectory-wise and stepwise policy optimization frameworks. Given two example group trajectories, Figure (b) illustrates trajectory-level and step-level grouping with their corresponding advantage estimations. Best viewed in color.

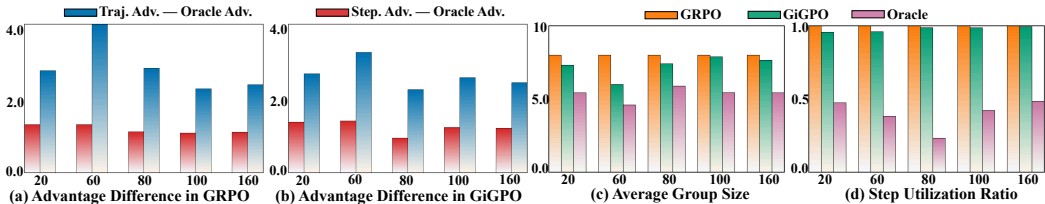

Figure 2: Statistics of GRPO and GiGPO. Figures (a) and (b) present the advantage differences relative to Oracle advantages for GRPO and GiGPO, respectively. Figures (c) and (d) report the average group size and the proportion of Oracle steps, respectively.

To address this issue, recent research has shifted toward the *stepwise* policy optimization framework (Feng et al., 2025b; Luo et al., 2025c; Chen et al., 2025b; Team, 2025; Yu et al., 2025b; Wang et al., 2025c), which treats each step within a rollout trajectory independently while leveraging a memory module to retain historical context. This design allows for flexible context management and highly scalable RL training. A comparison of the two frameworks is illustrated in Figure 1. (a). Building on the stepwise framework, group-based RL methods such as GRPO (Shao et al., 2024) can be adapted into stepwise group-based variants for long-horizon agentic tasks. Furthermore, to enable finer-grained credit assignment, GiGPO (Feng et al., 2025b) extends GRPO by estimating additional step-level advantages within groups where all steps share the same current state.

However, we identify a key issue in estimating stepwise group relative advantages: *historical context inconsistency*. This issue occurs when rollout steps share the same current state but differ in their historical contexts. As illustrated in Figure 1 (b), given two group trajectories $\tau_1$ and $\tau_2$, the step-level group at state $s_2$ (purple) contains steps with *inconsistent* historical contexts. Intuitively, the relative advantage of the selected actions should be computed under the same current state and the same historical context. When historical contexts vary, the estimated relative advantage can become *biased*, which in turn may degrade policy optimization.

To further explore this issue, we conduct a pilot empirical analysis. We introduce *Oracle* groups, in which all steps share not only the same current state but also identical historical contexts. During GRPO and GiGPO training, we track the group sizes, step counts, and estimated advantages of these Oracle groups, and compare them with trajectory-level and step-level advantage estimates. As shown in Figure 2 (a) and (b), both trajectory-level and step-level advantages exhibit notable estimation bias, with the bias at the trajectory level being substantially larger. These results indicate that historical context inconsistency can severely distort advantage estimation. A straightforward solution is to use only Oracle steps for policy optimization. However, as shown in Figures 2 (c) and (d), Oracle steps are generally scarce within trajectories (i.e., their ratio is low), making this approach inefficient. Moreover, the average group size of Oracle steps is small, which increases the variance of estimated advantages and undermines the stability of RL training.

To address the above challenges, in this paper, we propose Hierarchy-of-Groups Policy Optimization (HGPO), a novel RL training algorithm that introduces a better advantage estimator capable

of low-bias and balanced-variance. Specifically, HGPO is built on two key components: context-aware hierarchical grouping and adaptive weighting advantage estimation. First, within each rollout, HGPO groups steps that share the same current state and further assigns them to multiple hierarchical groups according to their historical contexts. This hierarchical structure captures advantages at different context depths, improving data utilization and reducing variance. Second, HGPO aggregates the group advantages using an adaptive weighting scheme: groups with more consistent historical contexts are assigned larger weights, thereby lowering estimation bias. In this way, HGPO produces more reliable stepwise advantage estimates for policy optimization. We evaluate HGPO on two challenging agentic benchmarks, ALFWorld and WebShop, using Qwen2.5-1.5B-Instruct and Qwen2.5-7B-Instruct. Results show that HGPO consistently outperforms existing baselines while maintaining the same GPU memory usage, using identical LLM rollouts, and incurring minimal additional time cost. Our main contributions are summarized as follows:

- ***Revealing historical context inconsistency.*** We reveal the issue of context inconsistency in stepwise group-based RL and empirically demonstrate that it introduces significant bias in advantage estimation, thereby degrading policy optimization.

- ***Proposing a novel policy optimization algorithm.*** We introduce Hierarchy-of-Groups Policy Optimization, which constructs hierarchical groups for each step based on historical context and adaptively aggregates their advantages.

- ***Achieving strong empirical performance.*** HGPO achieves state-of-the-art results on two challenging agentic benchmarks under the same computational constraints.

## 2 RELATED WORK

**LLM-based decision-making agents.** Large language models (LLMs) have been widely adopted as autonomous agents across diverse domains, including device control (Zhang & Zhang, 2024; Hong et al., 2024; Gur et al., 2024; Hu et al., 2024), code generation (Zhang et al., 2024b), game interaction (Wang et al., 2024a; Tan et al., 2025), and robotics (Zitkovich et al., 2023). Early approaches often relied on fixed pre-trained models guided by structured prompting, such as Re-Act (Yao et al., 2023) and Reflexion (Shinn et al., 2024), augmented with memory and retrieval mechanisms (Wang et al., 2024b; Tan et al., 2024) or tool integration (Schick et al., 2023; Xie et al., 2024; Zhang et al., 2024a). While such methods are simple and require no additional training, they remain limited in applicability to domain-specific tasks, largely due to the lack of specialized knowledge in the pre-training of the base models.

**Reinforcement learning for LLM-based agents.** Reinforcement learning (RL) (Sutton & Barto, 2018) has been central to adapting large language model (LLM) agents to dynamic and open-ended environments. Early work applied classic algorithms such as DQN (Mnih et al., 2015) to text games (Narasimhan et al., 2015), followed by value-based methods like PPO (Schulman et al., 2017) and AWR (Peng et al., 2019) in interactive domains including mobile control (Rawles et al., 2024), embodied tasks in ALFWorld (Shridhar et al., 2021), and card games (Brockman, 2016). More recent research has extended RL to web and application environments (Qian et al., 2025; Sun et al., 2025), with methods such as ArCHer (Zhou et al., 2024b), AgentQ (Putta et al., 2024), CoSo (Feng et al., 2025a), and LOOP (Chen et al., 2025a). In parallel, RL has also become integral to LLM training itself, with RLHF (Ziegler et al., 2019; Stiennon et al., 2020; Ouyang et al., 2022; Rafailov et al., 2024) aligning models with human preferences, and group-based RL algorithms emerging as scalable and efficient alternatives to PPO. Approaches such as GRPO (Shao et al., 2024), Dr. GRPO (Liu et al., 2025), Clip-Cov (Cui et al., 2025), GSPO (Zheng et al., 2025), and DAPO (Yu et al., 2025c) avoid value networks by estimating advantages over groups of samples. However, most of these methods are designed for single-turn interactions and thus struggle with context consistency in long-horizon agentic tasks.

**Long-horizon agentic reinforcement learning.** Long-horizon agentic RL (Laban et al., 2025; Zhang et al., 2025; Zhou et al., 2025a; Luo et al., 2025d; Wang et al., 2025a) extends LLMs from single-turn generation to multi-turn decision-making, where RL equips them with planning (Hao et al., 2023; Zhou et al., 2024a; Song et al., 2024), reasoning (Chu et al., 2025), and memory (Jin et al., 2024; Chhikara et al., 2025; Zhou et al., 2025b) capabilities for sustained interaction in dynamic environments. Applications span code generation (Jiang et al., 2024; Gehring et al., 2025; Jain et al., 2025; Chen et al., 2025c; Jin et al., 2025b), software en-

gineering (Wei et al., 2025b; Luo et al., 2025a; Shen et al., 2025; Wang et al., 2024c; Lin et al., 2025), and GUI interaction (Wei et al., 2025c; Lu et al., 2025; Luo et al., 2025b; Qin et al., 2025). Recent advances include long-horizon policy optimization frameworks (Wang et al., 2025d; Jin et al., 2025a) that optimize over multi-turn rollouts, and stepwise policy optimization methods (Feng et al., 2025b; Luo et al., 2025c; Chen et al., 2025b; Team, 2025) that treat each step independently while retaining history through memory modules. Yet, stepwise methods often suffer from context inconsistency across long horizons, limiting their effectiveness in complex agentic tasks.

## 3 PRELIMINARIES

**Problem setup of long-horizon agentic tasks.** Unlike single-turn tasks, long-horizon agentic tasks require an LLM agent to interact with the environment across multiple turns to accomplish a goal. Formally, given a task example $\boldsymbol{x} \in p(X)$, which typically includes a fixed task-related description, an LLM-based agent $\pi_\theta$ parameterized by $\theta$ observes an environment state $\boldsymbol{s}_t \in \mathcal{S}$ at each turn $t$ and generates a textual action $\boldsymbol{a}_t \in \mathcal{V}^n$, where $\mathcal{V}$ denotes the token vocabulary and $n$ is the maximum generation length. Here $t = (1, 2, \ldots, T)$, with $T$ being the maximum number of interaction turns. In this paper, we focus on the sparse delayed reward setting, where the environment provides a scalar reward $r_t \in \mathcal{R}$ only at the final step of a trajectory $\tau = \{(\boldsymbol{s}_1, \boldsymbol{a}_1), \ldots, (\boldsymbol{s}_T, \boldsymbol{a}_T)\}$.

**Trajectory-wise vs. stepwise policy optimization.** Conventional trajectory-wise policy optimization frameworks (Wang et al., 2025d; Jin et al., 2025a; Wang et al., 2025b; Yu et al., 2025a) typically concatenate the full interaction history of a rollout trajectory $\tau$ for policy optimization, i.e., $\pi_\theta(\boldsymbol{a}_t | \boldsymbol{s}_{0:t}, \boldsymbol{x})$. However, as the number of turns $T$ grows, the context length increases rapidly, which limits the scalability and feasibility of long-horizon RL training. In contrast, stepwise policy optimization frameworks (Feng et al., 2025b; Luo et al., 2025c; Chen et al., 2025b; Team, 2025) decouple the trajectory into individual steps while leveraging a memory module that maintains $K \ll T$ historical contexts. This memory module is updated with the latest $K$ interactions, keeping the prompt length relatively stable and enabling more scalable RL training.

**Group-based reinforcement learning.** Unlike PPO (Schulman et al., 2017), which estimates advantages using an additional value function, group-based reinforcement learning (RL) algorithms such as GRPO (Shao et al., 2024) compute advantages directly from the statistics of a sampled group of trajectories $G_\tau$. Specifically, GRPO was originally designed for single-turn tasks under a trajectory-wise policy optimization framework. To extend it to long-horizon tasks, we adapt it to the stepwise setting and calculate the trajectory-level advantage as:

$$A^T(\tau_i) = \left( R(\tau_i) - 1/|G_\tau| \sum\nolimits_{j \in G_\tau} R(\tau_j) \right) / \sigma_{G_\tau}, \tag{1}$$

where $\sigma_{G_\tau}$ denotes the standard deviation of rewards within the group $G_\tau$. This trajectory-level computation assigns the same advantage value to every step in trajectory $\tau_i$, thereby overlooking the finer credit assignment required within a trajectory. To address this limitation, one can instead adopt a step-level group relative advantage estimator (Feng et al., 2025b). Here, steps with identical current states $\tilde{\boldsymbol{s}}_i$ across all group trajectories are clustered into step-level groups $G_{\tilde{\boldsymbol{s}}_i}$, and their advantages are computed as:

$$A^S(\tilde{\boldsymbol{s}}_i) = \left( R(\tilde{\boldsymbol{s}}_i) - 1/|G_{\tilde{\boldsymbol{s}}_i}| \sum\nolimits_{j \in G_{\tilde{\boldsymbol{s}}_i}} R(\tilde{\boldsymbol{s}}_j) \right) / \sigma_{G_{\tilde{\boldsymbol{s}}_i}}. \tag{2}$$

Compared to Eq. (1), the step-level estimator in Eq. (2) provides more fine-grained and effective credit assignment across steps within the same trajectory.

## 4 TRAINING AGENTS WITH HGPO FOR LONG-HORIZON AGENTIC TASKS

### 4.1 THE ISSUE OF HISTORICAL CONTEXT INCONSISTENCY

Originally, group relative advantage estimation (Shao et al., 2024) compares the relative advantages of different group responses generated from the same prompt. However, in the stepwise policy optimization setting shown in Figure 1, even for a fixed task in the environment, rollout steps within a step-level anchor group (i.e., steps that share the same current state) may still have *distinct historical*

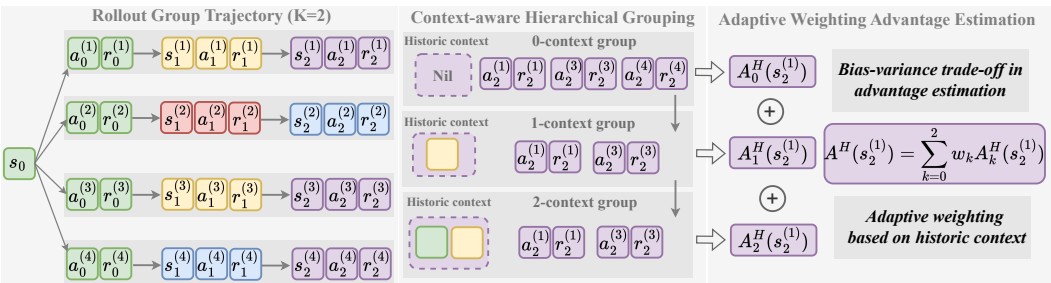

Figure 3: Overview of HGPO. The LLM-based agent interacts with a set of environments initialized from the same state $s_0$, producing four group trajectories (states with the same color are identical). HGPO comprises two key components: context-aware hierarchical grouping and adaptive weighted advantage computation. For illustration, consider the state $s_2$ (purple). First, HGPO assigns $s_2$ into three hierarchical groups according to its historical contexts. Then, it computes the final advantage estimate by adaptively aggregating the weighted advantages from these groups.

*contexts* in their memory modules. As a result, the effective prompts for these step-level groups can differ, which may bias the estimated advantages. Ideally, Oracle steps, i.e., those sharing the same prompt, including both the current state and the historical context, would yield the most accurate advantage estimates for policy optimization. A straightforward approach is therefore to perform policy optimization using only Oracle steps. In practice, however, Figure 2 shows that Oracle steps are rare in rollouts, and their group sizes are typically small, which can destabilize training. Motivated by these challenges, we propose leveraging a hierarchy-of-groups structure to obtain more accurate advantage estimates, reducing bias while keeping variance low.

## 4.2 Hierarchy-of-Groups Policy Optimization

In this subsection, we introduce HGPO as shown in Figure 3, consisting of context-aware hierarchical grouping and adaptive weighting advantage estimation.

**Context-aware hierarchical grouping.** We begin by introducing *context-aware hierarchical grouping*, which organizes steps into multi-level groups according to their historical contexts. The key intuition is that the advantage of each step should be evaluated relative to different historical contexts to obtain more accurate estimates. Specifically, we first group together steps that share the same current state, and then, within each group, we construct multiple hierarchical groups based on the consistency of their historical contexts. Steps with longer common histories are assigned to higher-level hierarchical groups. This hierarchy-of-groups structure enables more fine-grained comparisons and brings two main benefits: (i) it improves step utilization for advantage estimation, and (ii) it reduces the variance of estimated advantages.

Formally, let the $i$-th trajectory be $\tau_i = \{(s_1^{(i)}, a_1^{(i)}), (s_2^{(i)}, a_2^{(i)}), \ldots, (s_T^{(i)}, a_T^{(i)})\}$, and let $K$ denote the maximum context length. We define a $k$-step context operator for the $t$-th step as:

$$\mathcal{C}_k(s_t^{(i)}) = \begin{cases} \left(s_{t-k}^{(i)}, s_{t-k+1}^{(i)}, \cdots, s_t^{(i)}\right), t \geq k, \\ \left(s_0^{(i)}, s_1^{(i)}, \cdots, s_t^{(i)}\right), t < k, \end{cases} \tag{3}$$

where $k \in [0, K]$. This operator returns the $k$ historical states preceding the current state. Based on this operator, we define the $k$-th hierarchical group for the $t$-th step as:

$$G_k^H(s_t^{(i)}) = \left\{(j, n) \in \mathcal{I} : \mathcal{C}_k(s_t^{(i)}) = \mathcal{C}_k(s_n^{(j)})\right\}, \tag{4}$$

where the index set $\mathcal{I} = \{(i, t) \mid 1 \leq i \leq N,, 1 \leq t \leq T\}$. Considering all hierarchical groups, the resulting hierarchy-of-groups structure satisfies:

$$G_0^H(s_t^{(i)}) \supseteq G_1^H(s_t^{(i)}) \supseteq \cdots \supseteq G_K^H(s_t^{(i)}), \qquad \left|G_0^H(s_t^{(i)})\right| \geq \cdots \geq \left|G_K^H(s_t^{(i)})\right|. \tag{5}$$

When $K = 0$, the hierarchy-of-groups degenerates to the step-level grouping $G_0^H(s_t^{(i)})$ used in (Feng et al., 2025b). Importantly, the entire context-aware hierarchical grouping procedure operates fully offline: it requires only hashmap lookups over existing rollouts, without relying on additional models or extra data collection.

**Adaptive weighting advantage estimation.** Intuitively, higher-level hierarchical groups yield more accurate advantage comparisons since they incorporate richer historical context. Building on this insight, we introduce an adaptive weighting scheme that integrates information across all hierarchical groups with appropriately assigned weights, thereby enabling stable and efficient estimation of group-relative advantages. Formally, the advantage estimation for the $k$-th hierarchical group is defined as:

$$A_k^H(\boldsymbol{s}_t^{(i)}) = \left( R(\boldsymbol{s}_t^{(i)}) - 1/|G_k^H| \sum_{(j,n)\in G_k^H} R(\boldsymbol{s}_n^{(j)}) \right) / \sigma_{G_k^H}. \tag{6}$$

Finally, the advantage aggregated from $K$ hierarchical groups is denoted by:

$$A^H(\boldsymbol{s}_t^{(i)}) = \sum_{k=0}^{K} \boldsymbol{w}_k A_k^H(\boldsymbol{s}_t^{(i)}), \tag{7}$$

where the adaptive weight $\boldsymbol{w}_k = \frac{(k+1)^\alpha}{\sum_k (k+1)^\alpha}$ ($\alpha \geq 0$). It is worth noting that Eq. (7) fuses advantage information along the hierarchy-of-groups in Eq. (5): higher-level groups are preferred due to stronger context consistency. Besides, for each step $(\boldsymbol{s}_t^{(i)}, \boldsymbol{a}_t^{(i)})$ we compute its stepwise reward $r_t^{(i)} = \sum_{j=t}^{T} \gamma^{j-t} r_j^{(i)}$ (Feng et al., 2025b), where $\gamma \in (0, 1]$ is the discount factor. In this way, we can obtain a stepwise reward for each step in the trajectory.

**The objective for policy optimization.** The policy optimization objective of HGPO is:

$$\mathcal{J}_{\text{HGPO}}(\theta) = \mathbb{E}\left[ \frac{1}{NT} \sum_{i=1}^{N} \sum_{t=1}^{T} \min\left( \rho_\theta(\boldsymbol{a}_t^{(i)}) A^H(\boldsymbol{s}_t^{(i)}), \, \text{clip}\left( \rho_\theta(\boldsymbol{a}_t^{(i)}), 1 \pm \epsilon \right) A^H(\boldsymbol{s}_t^{(i)}) \right) \right]$$
$$- \beta \mathbb{D}_{\text{KL}}\left( \pi_\theta(\cdot \mid x) \,\|\, \pi_{\text{ref}}(\cdot \mid x) \right), \tag{8}$$

where $\rho_\theta(\boldsymbol{a}_t^{(i)}) = \frac{\pi_\theta(\boldsymbol{a}_t^{(i)}|\boldsymbol{s}_t^{(i)},x)}{\pi_{\theta_{\text{old}}}(\boldsymbol{a}_t^{(i)}|\boldsymbol{s}_t^{(i)},x)}$ is the importance sampling ratio, $\beta$ controls the strength of the KL penalty. The pseudo-code is shown in Algorithm 1 of Appendix A.

**Proposition 4.1 (Bias-variance trade-off in HGPO)** *Let $b_k$ and $v_k$ denote the bias and variance of the estimated advantage $A_k^H$ within the $k$-th group $G_k^H$. Based on the following conditions: (1)* Bias *satisfies, i.e., $B_T \geq b_0 \geq (b_1, \cdots, b_{K-1}) \geq b_K \geq 0$; (2)* Variance *satisfies, i.e., $v_0 \leq (v_1, \cdots, v_{K-1}) \leq v_K \leq V_T$, the bias and variance of the estimator $A^H$ are*

$$Bias[A^H] = Bias\left[ \sum_{k=0}^{K} w_k A_k^H \right] = \sum_{k=0}^{K} w_k b_k,$$

$$Var[A^H] = Var\left[ \sum_{k=0}^{K} w_k A_k^H \right] = \sum_{k=0}^{K} w_k^2 Var[A_k^H] = \sum_{k=0}^{K} w_k^2 v_k.$$

*Furthermore, the bias and variance of the advantage estimator in HGPO satisfy that*

$$b_K \leq Bias[A^H] \leq b_0 \leq B_T,$$

$$\frac{v_0}{K+1} \leq Var[A^H] \leq v_K \leq V_T,$$

*where $B_T, b_0, b_K$, and $V_T, v_0, v_K$ denote the bias and variance of the trajectory-level, step-level, and Oracle advantage, respectively. Overall, the bias and variance of the HGPO advantage estimator interpolates between the step-level ($k = 0$) and Oracle ($k = K$) estimators, thereby achieving a better trade-off.* Proof and more details are provided in Appendix B.

## 5 EXPERIMENTS

### 5.1 EXPERIMENT SETUP

**Agentic benchmarks.** We train the LLM agents on two challenging benchmarks: ALF-World (Shridhar et al., 2021) and WebShop (Yao et al., 2022), which are designed to assess the ability of LLM agents to perform multi-step decision-making. The details are shown in Appendix C.2.

Table 1: Performance comparison on ALFWorld and WebShop. For ALFWorld, we report the overall success rate (↑) for both *in-distribution* (In-Success) and *out-of-distribution* tasks (Out-Success). For WebShop, we report the average task score (↑) and the average task success rate (↑). Most results are averaged over 3 random seeds during testing. The best results are highlighted in bold.

| Model | Type | Method | ALFWorld | | WebShop | |
|---|---|---|---|---|---|---|
| | | | In-Success | Out-Success | Task Scores | Task Success Rates |
| Closed | Prompting | GPT-4o | 48.0 | 46.0 | 31.8 | 23.7 |
| | Prompting | Gemini-2.5-Pro | 60.3 | 50.5 | 42.5 | 35.9 |
| Qwen2.5-1.5B-Instruct | Prompting | Qwen2.5 | 4.1 | - | 23.1 | 5.2 |
| | Prompting | ReAct | 12.8 | - | 40.1 | 11.3 |
| | Prompting | Reflexion | 21.8 | - | 55.8 | 21.9 |
| | RL Training | PPO (with critic) | $54.4_{\pm 3.1}$ | - | $73.8_{\pm 3.0}$ | $51.5_{\pm 2.9}$ |
| | RL Training | RLOO | $69.7_{\pm 2.5}$ | $68.7_{\pm 10.7}$ | $73.9_{\pm 5.6}$ | $52.1_{\pm 6.7}$ |
| | RL Training | GRPO | $72.8_{\pm 3.6}$ | $70.1_{\pm 2.5}$ | $75.8_{\pm 3.5}$ | $56.8_{\pm 3.8}$ |
| | RL Training | GiGPO ($K$=2) | $90.16_{\pm 0.22}$ | $84.76_{\pm 2.83}$ | $84.95_{\pm 2.57}$ | $66.53_{\pm 1.92}$ |
| | RL Training | **HGPO** ($K$=2) | $\mathbf{92.77_{\pm 1.08}}$ | $\mathbf{90.16_{\pm 0.78}}$ | $\mathbf{85.56_{\pm 2.86}}$ | $\mathbf{71.54_{\pm 4.00}}$ |
| | RL Training | GiGPO ($K$=4) | $93.29_{\pm 1.07}$ | $91.53_{\pm 1.99}$ | $86.80_{\pm 1.60}$ | $73.24_{\pm 2.25}$ |
| | RL Training | **HGPO** ($K$=4) | $\mathbf{94.85_{\pm 0.92}}$ | $\mathbf{92.12_{\pm 1.63}}$ | $\mathbf{90.64_{\pm 1.05}}$ | $\mathbf{78.12_{\pm 2.06}}$ |
| Qwen2.5-7B-Instruct | Prompting | Qwen2.5 | 14.8 | - | 26.4 | 7.8 |
| | Prompting | ReAct | 31.2 | - | 46.2 | 19.5 |
| | Prompting | Reflexion | 42.7 | - | 58.1 | 28.8 |
| | RL Training | PPO (with critic) | $77.08_{\pm 1.12}$ | $76.23_{\pm 1.46}$ | $81.4_{\pm 3.1}$ | $68.7_{\pm 5.1}$ |
| | RL Training | RLOO | $77.86_{\pm 0.03}$ | $73.95_{\pm 0.05}$ | $80.3_{\pm 3.2}$ | $65.7_{\pm 4.0}$ |
| | RL Training | GRPO | $78.64_{\pm 0.73}$ | $76.82_{\pm 1.47}$ | $79.3_{\pm 2.8}$ | $66.1_{\pm 3.7}$ |
| | RL Training | GiGPO ($K$=2) | $93.29_{\pm 0.40}$ | $92.18_{\pm 0.39}$ | $88.93_{\pm 1.49}$ | $77.60_{\pm 1.68}$ |
| | RL Training | **HGPO** ($K$=2) | $\mathbf{95.44_{\pm 0.62}}$ | $\underline{92.05_{\pm 0.22}}$ | $\mathbf{88.96_{\pm 1.04}}$ | $\mathbf{78.51_{\pm 1.40}}$ |
| | RL Training | GiGPO ($K$=4) | $95.63_{\pm 0.49}$ | $95.18_{\pm 0.98}$ | $89.51_{\pm 0.59}$ | $77.79_{\pm 0.92}$ |
| | RL Training | **HGPO** ($K$=4) | $\mathbf{95.96_{\pm 0.49}}$ | $\underline{94.87_{\pm 1.24}}$ | $\underline{88.49_{\pm 0.41}}$ | $\mathbf{79.29_{\pm 1.17}}$ |

**Comparing methods.** We compare HGPO with many competitive baselines: (1) Closed-source LLMs: GPT-4o (Achiam et al., 2023) and Gemini-2.5-Pro (Team et al., 2023). (2) Prompting agents: ReAct (Yao et al., 2023) and Reflexion (Shinn et al., 2024). (3) RL training methods: PPO (Schulman et al., 2017), RLOO (Kool et al., 2019; Ahmadian et al., 2024), GRPO (Shao et al., 2024), and GiGPO (Feng et al., 2025b). The details are shown in Appendix C.1.

**Implementation details.** We adopt Qwen2.5-1.5B-Instruct and Qwen2.5-7B-Instruct (Yang et al., 2024) as our base models. For fairness, all RL training methods share the same hyperparameter configurations. Specifically, the rollout group size $N$ in group-based RL methods is set to 8. Each LLM agent is prompted to first generate a chain-of-thought (Wei et al., 2022) enclosed within <think> </think> tags, followed by the action enclosed within <action> </action> tags. For HGPO, the weighting coefficient $\alpha$ in Eq. (7) is set to 1, and we omit groups with zero advantage in Eq. 7 for adaptive weighting. For evaluation, we set three different random seeds and report the mean and standard deviation of the performance. The max step is set to 50 and 30 for ALFWorld and Web-Shop, respectively. Full training setups and hyperparameter details are provided in Appendix C.3.

## 5.2 EXPERIMENTAL RESULTS

***HGPO achieves the best overall results.*** Table 1 shows a pronounced gap between prompting and RL training methods. On both ALFWorld and WebShop, all RL-trained methods substantially outperform the prompting baselines (Qwen2.5, ReAct, Reflexion), confirming that RL training is crucial for these long-horizon tasks. With Qwen2.5-1.5B-Instruct, HGPO consistently improves over GiGPO by an average of 4.01% on ALFWorld ($K = 2$), 1.08% on ALFWorld ($K = 4$), 2.81% on WebShop ($K = 2$), and 4.36% on WebShop ($K = 4$). With Qwen2.5-7B-Instruct, HGPO remains superior, achieving up to 95.96% in-distribution success rate on ALFWorld and 79.29% task success rate on WebShop. Moreover, all baseline methods experience significant performance degradation on out-of-distribution tasks in ALFWorld. Notably, HGPO maintains superior performance

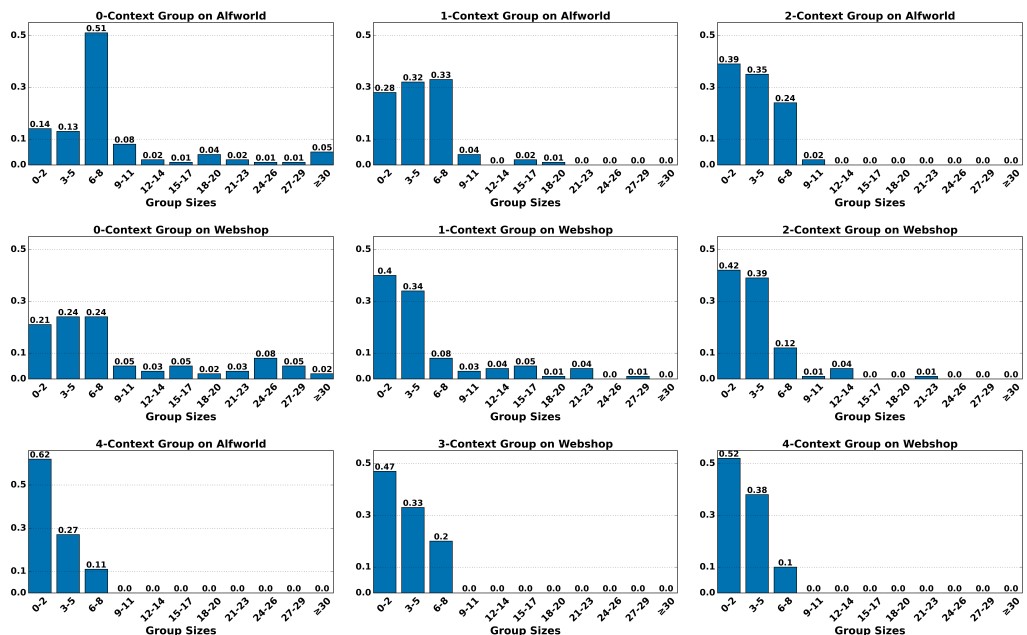

Figure 4: Distributions of hierarchical group sizes on ALFWorld and WebShop using Qwen2.5-1.5B-Instruct. "0/1/2/3/4-Context" indicates different hierarchical groups. The first two rows correspond to $K = 2$, and the last row corresponds to $K = 4$. The y-axis denotes the proportion.

with less degradation compared to GiGPO. This observation suggests that context inconsistency can severely impair policy generalization, while HGPO's hierarchical grouping mechanism provides robust and stable advantage estimation, enabling improved generalization to unseen tasks. Overall, HGPO achieves the strongest performance across settings.

***HGPO brings larger gains on small models.*** Table 1 shows that HGPO achieves larger improvement with Qwen2.5-1.5B-Instruct than with Qwen2.5-7B-Instruct, with an average gain of 3.41% vs. 0.74% for $K = 2$ and 2.72% vs. 0.13% for $K = 4$. We explain that Qwen2.5-1.5B-Instruct tends to generate longer and more redundant steps during rollout due to its limited agentic capability, which can introduce larger bias in advantage estimation. In this case, using hierarchical groups becomes more important for refining the advantage estimate and stabilizing policy optimization. By contrast, Qwen2.5-7B-Instruct often solves tasks with fewer but more accurate steps, leading to a smaller advantage-estimation bias. Consequently, the additional benefit from hierarchical grouping is more modest (though still cost-effective). Overall, these results suggest that HGPO is particularly well-suited to challenging long-horizon agentic tasks where rollouts are lengthy, and advantage estimates are substantially biased.

***HGPO consistently achieves superior performance with different*** $K$***.*** We also observe that increasing $K$ improves the performance of both GiGPO and HGPO. As $K$ grows, the memory module can retain richer historical context, allowing the policy to make better use of past information when selecting actions. HGPO benefits from this larger memory as well, demonstrating that it scales effectively with the memory size.

## 5.3 FURTHER ANALYSIS

**Distribution of hierarchical group sizes.** Figure 4 presents the distribution of hierarchical group sizes on ALFWorld and WebShop with Qwen2.5-1.5B-Instruct at the 160th epoch. "0/1/2/3/4-Context" denotes steps that share 0/1/2/3/4 identical historical contexts. We find that 0-context groups have a larger fraction of large groups than 1-context and 2-context groups, since they ignore history. As $K$ increases, the mass shifts away from large groups toward smaller ones, and small groups become more common. This pattern suggests that Oracle steps sharing the same histori-

Table 2: Step utilization ratio on ALFWorld and WebShop using Qwen2.5-1.5B-Instruct. "Context" is abbreviated as "C", indicating different hierarchical groups.

| Dataset | $K$ | 0-C | 1-C | 2-C | 3-C | 4-C |
|---------|-----|-----|-----|-----|-----|-----|
| ALFWorld | 2 | 0.97 | 0.75 | 0.52 | - | - |
|          | 4 | 0.98 | 0.77 | 0.54 | 0.34 | 0.19 |
| WebShop | 2 | 0.92 | 0.64 | 0.44 | - | - |
|         | 4 | 0.90 | 0.59 | 0.40 | 0.21 | 0.09 |

Table 3: Time cost (s) and Peak memory (MB) for hashing lookups at varying training epochs.

| Metric | Epoch | 40 | 80 | 120 | 160 |
|--------|-------|-----|-----|-----|-----|
| Time | Common | 297.8 | 288.6 | 284.7 | 282.5 |
|      | GRPO | 0.048 | 0.027 | 0.018 | 0.013 |
|      | GiGPO | 0.126 | 0.080 | 0.055 | 0.034 |
|      | HGPO | 0.693 | 0.601 | 0.456 | 0.246 |
| Mem. | GiGPO | 0.1035 | 0.0967 | 0.0605 | 0.0391 |
|      | HGPO | 0.1393 | 0.1229 | 0.1078 | 0.0406 |

cal context typically form small groups, which can increase the variance of advantage estimation. Additional results are provided in Appendix D.3.

**Step utilization ratio.** Table 2 reports the average proportion of steps allocated to different context groups per rollout in ALFWorld and WebShop using Qwen2.5-1.5B-Instruct. The results show that nearly all steps fall into 0-context groups, except for a small fraction corresponding to unique states (appearing only once in a group). As the number of historical contexts increases, the utilization ratio steadily decreases, since fewer steps can be aggregated into higher-level groups. This finding highlights the challenge posed by the scarcity of Oracle steps.

**The computational budget analysis of HGPO.** We show a detailed computational budget analysis of HGPO. Specifically, HGPO shares the same core architecture as GRPO and GiGPO. The common computational components include multi-turn rollouts, computation of old and reference probabilities, and clipped policy updates. All methods are critic-free and operate with a single actor LLM, resulting in identical GPU memory usage and LLM rollout costs. The primary addition of HGPO lies in advantage estimation. To evaluate its computational cost, we measured the per-iteration training time using Qwen2.5-1.5B-Instruct on ALFWorld. From Table 3, we can summarize three key observations. (i) As the number of training epochs increases, both the time cost and peak memory usage consistently decrease, since the rollout steps become fewer when the agent learns to accomplish the tasks with fewer steps. (ii) HGPO introduces an average additional time cost of approximately 0.425 s and 0.472 s compared with GRPO and GiGPO, respectively, which corresponds to less than 0.001% of the total execution time. These results demonstrate that HGPO maintains computational efficiency comparable to that of GRPO and GiGPO. (iii) HGPO only causes a slight increase in the peak memory usage due to the additional hashing lookups. Overall, HGPO preserves the high computational and memory efficiency of GRPO and GiGPO.

## 5.4 PARAMETER ANALYSIS

Table 4: Parameter analysis on the effects of different values of $\alpha$ on ALFWorld and WebShop using Qwen2.5-1.5B-Instruct.

| $\alpha$ | $K$ | ALFWorld | | WebShop | |
|----------|-----|----------|--------|---------|---------|
|          |     | In-Suc. | Out-Suc. | Task Sco. | Task Suc. |
| 0 | 2 | $94.33_{\pm1.08}$ | $90.03_{\pm2.58}$ | $88.40_{\pm2.20}$ | $73.95_{\pm4.21}$ |
|   | 4 | $92.12_{\pm0.40}$ | $89.58_{\pm1.00}$ | $87.96_{\pm2.48}$ | $73.82_{\pm3.26}$ |
| 1 | 2 | $92.77_{\pm1.08}$ | $90.16_{\pm0.78}$ | $85.56_{\pm2.86}$ | $71.54_{\pm4.00}$ |
|   | 4 | $94.85_{\pm0.92}$ | $92.12_{\pm1.63}$ | $90.64_{\pm1.05}$ | $78.12_{\pm2.06}$ |
| 2 | 2 | $90.36_{\pm0.96}$ | $86.91_{\pm2.03}$ | $88.29_{\pm2.88}$ | $72.00_{\pm3.03}$ |
|   | 4 | $93.55_{\pm0.51}$ | $86.78_{\pm2.26}$ | $88.28_{\pm2.24}$ | $75.39_{\pm2.95}$ |

Here, we study the effect of different values of $\alpha$ in Eq. (7). Notably, $\alpha$ controls how sharp the weight distribution is, i.e., a larger $\alpha$ puts more weight on high-level groups. The experimental results are shown in Table 4. We can see that when $K = 2$, HGPO with $\alpha = 0$ achieves the comparatively better performance on both ALFWorld and WebShop, while when $K = 4$, the performance of HGPO with $\alpha = 0$ decreases. In contrast, the performance of HGPO with $\alpha = 1$ and $\alpha = 2$ both increases when $K$ increases from 2 to 4. These observations indicate that as $K$ increases, up-weighting the advantage estimate from higher-level groups improves the performance. This is because, as $K$ increases, higher-level groups can provide more accurate advantage estimation. On the other hand, HGPO with $\alpha = 2$ slightly drops the performance on both ALFWorld and WebShop, compared with that with $\alpha = 1$. This is because, although the bias of advantage estimation in higher-level groups is relatively low, the variance could be high due to the small sample size in the group. Hence, ex-

cessively emphasizing the advantage estimation in higher-level groups is not always optimal, and a bias-variance trade-off exists in hierarchical groups. Overall, we use $\alpha = 1$ as the default. In the future, it is also interesting to develop a better adaptive weighting scheme based on the uncertainty of advantage estimation in hierarchical groups.

## 5.5 ABLATION STUDY

Table 5: Ablation study on ALFWorld and Web-Shop using Qwen2.5-1.5B-Instruct.

| Ablation | $K$ | ALFWorld | | WebShop | |
|---|---|---|---|---|---|
| | | In-Suc. | Out-Suc. | Task Sco. | Task Suc. |
| HGPO | 2 | $92.77_{\pm1.08}$ | $90.16_{\pm0.78}$ | $85.56_{\pm2.86}$ | $71.54_{\pm4.00}$ |
| | 4 | $94.85_{\pm0.92}$ | $92.12_{\pm1.63}$ | $90.64_{\pm1.05}$ | $78.12_{\pm2.06}$ |
| w/o $G_{1:K}^H$ | 2 | $90.88_{\pm1.30}$ | $85.15_{\pm1.73}$ | $87.89_{\pm0.86}$ | $73.24_{\pm1.47}$ |
| | 4 | $91.14_{\pm1.07}$ | $88.80_{\pm3.21}$ | $88.83_{\pm2.68}$ | $74.73_{\pm3.62}$ |
| w/o Ada. $w_k$ | 2 | $94.33_{\pm1.08}$ | $90.03_{\pm2.58}$ | $88.40_{\pm2.20}$ | $73.95_{\pm4.21}$ |
| | 4 | $92.12_{\pm0.40}$ | $89.58_{\pm1.00}$ | $87.96_{\pm2.48}$ | $73.82_{\pm3.26}$ |
| w Eq. (1) | 2 | $88.21_{\pm1.95}$ | $84.76_{\pm2.05}$ | $89.36_{\pm1.27}$ | $76.82_{\pm2.25}$ |
| | 4 | $91.27_{\pm1.17}$ | $90.55_{\pm0.29}$ | $87.63_{\pm0.80}$ | $73.50_{\pm1.85}$ |

In this section, we conduct an ablation study to evaluate the effectiveness of each component in HGPO. As shown in Table 5, "w/o $G_{1:K}^H$" denotes the setting where hierarchical grouping is removed, and only the original step-level group $G_0^H$ is used to compute relative advantages for policy optimization. This configuration results in the remarkable performance degradation on ALFWorld when $K = 2/4$, i.e., about 2.8% of in-distribution success rates and 5.8% of out-of-distribution success rates. This observation directly validates that directly using the current state of steps for grouping could lead to biased advantage estimation and degrade the policy optimization, and our proposed hierarchy-of-groups structures effectively refine the advantage estimation. A similar result can also be observed on WebShop, except for $K = 2$, which results in a little performance improvement (when $\alpha = 1$). Second, "w/o Ada. $w_k$" refers to replacing adaptive weighting with uniform weights, i.e., $\alpha = 0$ in Eq. (7). We can see that the performance slightly increases on both ALFWorld and WebShop when $K = 2$ and decreases when $K = 4$. This is because when $K = 2$, the relative advantage bias of lower-level hierarchical groups is lower than when $K = 4$, and thus the average weighting can also be beneficial for advantage estimation. Once $K$ increases, the advantage bias of lower-level hierarchical groups could be increased and degrade the policy optimization. Hence, the adaptive weighting mechanism is important for advantage estimation, and requires no complex hyperparameter tuning and remains scalable across different lengths of historical contexts. Third, "w Eq. (1)" indicates using the additional trajectory-level advantage in Eq. (1) for the final advantage estimation in Eq. (7). We can see that the performance of almost all decreases on ALFWorld and WebShop, except for $K = 2$ on WebShop. This may be because the trajectory-level advantage is generally highly biased and low-varied, which could not provide effective information for policy optimization in most cases. Besides, we also conduct experiments that only use Oracle steps for policy optimization, but failed. Overall, we validate the effectiveness of each component in HGPO.

## 6 CONCLUSION

In this paper, we propose HGPO, a novel group-based reinforcement learning algorithm designed to mitigate historical context inconsistency in long-horizon agentic tasks. Specifically, HGPO introduces context-aware hierarchical grouping and adaptive weighting advantage estimation, which enables a better advantage estimate for policy optimization. Empirical results on two complex environments, ALFWorld and WebShop, show that HGPO substantially outperforms both prompt-based agents and prior RL approaches. In the future, an interesting direction is to explore the structure of hierarchy-of-groups on advanced agents with summarized memory. HGPO directly divides the hierarchical structure based on the raw, divisible historical contexts according to the historical steps. Currently, many advanced agents tend to summarize the historical contexts into the memory module. In this case, the straightforward division of hierarchical groups is intractable, and thus it is necessary to explore other ways for hierarchical grouping, e.g., the embedding similarity of the memory. Besides, it is also interesting to explore a totally adaptive weighting scheme for advantage aggregation from hierarchical groups by considering the uncertainty of the advantage estimate in each hierarchical group.

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

## A    ALGORITHM

---

**Algorithm 1** The pseudo-code of HGPO

---

1: **Require:** Initial policy $\pi_{\theta_{\text{old}}}$, task distribution $p(X)$, discount factor $\gamma$, weighting $\omega$, clipping
   parameter $\epsilon$, KL penalty $\beta$, group size $N$, the length of historical context $K$, parameter $\alpha$
2: **for** each training iteration **do**
3:     Update the old policy model: $\theta_{\text{old}} \leftarrow \theta$
4:     // Multi-step rollout phase
5:     Sample task $x \sim p(X)$ and initialize $N$ identical environments
6:     **for** $t = 1$ to $T$ **do**
7:         Sample actions $\left\{ \boldsymbol{a}_t^{(i)} \sim \pi_{\theta_{\text{old}}}(\cdot \mid \boldsymbol{s}_t^{(i)}, x) \right\}_{i=1}^N$
8:         Execute actions, observe rewards $\{r_t^{(i)}\}_{i=1}^N$ and next state $\{\boldsymbol{s}_{t+1}^{(i)}\}_{i=1}^N$
9:     **end for**
10:    // Grouping phase
11:    *Context-aware hierarchical grouping by Eq. (5)*
12:    // Advantage computation phase
13:    *Compute multiple advantages within each group by Eq. (7)*
14:    // Policy update phase
15:    Update policy $\theta$ by maximizing objective $\mathcal{J}_{\text{HGPO}}(\theta)$
16: **end for**

---

## B    MORE DETAILS AND PROOF FOR THEOREM

Table 6: Overall comparison of three different advantage estimators.

| Type | Advantage estimation | Granularity | Bias | Variance |
|---|---|---|---|---|
| Trajectory-level | $A^T(\tau_i) = \left( R(\tau_i) - 1/\lvert G_\tau\rvert \sum_{j\in G_\tau} R(\tau_i) \right)/\sigma_{G_\tau}$ | Coarse-grained | $B_T$ | $V_T$ |
| Step-level | $A^S(\tilde{s}_i) = \left( R(\tilde{s}_i) - 1/\lvert G_{\tilde{s}_i}\rvert \sum_{j\in G_{\tilde{s}_i}} R(\tilde{s}_j) \right)/\sigma_{G_{\tilde{s}_i}}$ | Fine-grained | $b_0$ | $v_0$ |
| Hierarchy-of-Groups | $A^H(\boldsymbol{s}_t^{(i)}) = \sum_{k=0}^K \boldsymbol{w}_k A_k^H(\boldsymbol{s}_t^{(i)})$ | *Fine-grained* | $\sum_{k=0}^K w_k b_k \downarrow$ | $\sum_{k=0}^K w_k^2 v_k \downarrow$ |

Here, we provide more details of Proposition 4.1. Let $A_k^H$ denote the advantage estimator for the $k$-th hierarchical group. Let $b_k$ and $v_k$ denote the bias and variance of the estimated advantage $A_k^H$ within the $k$-th group $G_k^H$. The definition of bias is $b_k = \text{Bias}[A] = A - A^*$ where $A^*$ is the unknown true advantage. We make the following conditions:

(1) *Bias satisfies*:

$$B_T \geq b_0 \geq (b_1, b_2, \cdots, b_{K-1}) \geq b_K \geq 0, \quad b_k = \text{Bias}[A_k^H],$$

(2) *Variance satisfies*:

$$v_0 \leq (v_1, v_2, \cdots, v_{K-1}) \leq v_K \leq V_T, \quad v_k = \text{Var}[A_k^H],$$

where $B_T$ and $V_T$ denote the bias and variance of trajectory-level advantage estimation, and $b_0$ and $v_0$ represent those of step-level estimation. We now justify the assumptions. First, the number of trajectories in a group is generally smaller (set to 8 in our experiments) than the step-level group size, which leads to higher bias and variance in trajectory-level estimation. Second, as $K$ increases, the group size of $G_k^H$ decreases, which can result in higher variance.

**Bias and variance of HGPO.**    Recall the advantage aggregation in Eq. (5) and Eq. (7):

$$A^H = \sum_{k=0}^K w_k A_k^H$$

with group weights $w_k \geq 0$ satisfying $\sum_{k=0}^K w_k = 1$. We first analyze the bias. Define $b_k \triangleq \mathbb{E}[A_k^H] - A^*$ and $\text{Bias}[X] \triangleq \mathbb{E}[X] - A^*$, where $A^*$ denotes the target (unbiased) advantage. Then

$$\text{Bias}[A^H] = \mathbb{E}\left[\sum_{k=0}^K w_k A_k^H\right] - A^* = \sum_{k=0}^K w_k \left(\mathbb{E}[A_k^H] - A^*\right) = \sum_{k=0}^K w_k b_k.$$

Since $b_0 \geq b_1 \geq \cdots \geq b_K$ and $\sum_{k=0}^{K} w_k = 1$, it follows that

$$b_K = \sum\nolimits_{k=0}^{K} w_k b_K \leq \sum\nolimits_{k=0}^{K} w_k b_k = \text{Bias}[A^H] \leq \sum\nolimits_{k=0}^{K} w_k b_0 = b_0 \leq B_T.$$

Hence, HGPO trades off the bias between the step-level estimator ($k = 0$) and the oracle estimator ($k = K$). Correspondingly, for the variance, assume $\text{Cov}(A_k^H, A_{k'}^H) = 0$ for $k \neq k'$. Let $v_k \triangleq \text{Var}[A_k^H]$, then

$$\text{Var}[A^H] = \text{Var}\left[\sum_{k=0}^{K} w_k A_k^H\right] = \sum_{k=0}^{K} w_k^2 \text{Var}[A_k^H] = \sum_{k=0}^{K} w_k^2 v_k.$$

Since $v_0 \leq v_1 \leq \cdots \leq v_K$, then

$$v_0 \sum_{k=0}^{K} w_k^2 \leq \text{Var}[A^H] \leq v_K \sum_{k=0}^{K} w_k^2 \leq v_K.$$

Moreover, by Cauchy–Schwarz,

$$\frac{1}{K+1} \leq \sum_{k=0}^{K} w_k^2 \leq 1,$$

hence we can obtain:

$$\frac{v_0}{K+1} \leq \text{Var}[A^H] \leq v_K.$$

In summary, HGPO interpolates between the step-level ($k = 0$) and oracle ($k = K$) estimators in bias and variance, thereby achieving a better trade-off. It is also interesting to explore tighter bounds on the bias and variance in HGPO.

## C  EXPERIMENT DETAILS

### C.1  COMPARING METHODS

- *GPT-4o:* A closed-source, large-scale LLM used as a baseline for multi-turn agentic tasks (Achiam et al., 2023).
- *Gemini-2.5-Pro:* Another closed-source LLM, comparable in scale and capability to GPT-4o (Team et al., 2023).
- *ReAct:* A prompting-based agent that integrates reasoning and acting in an interleaved chain-of-thought framework (Yao et al., 2023).
- *Reflexion:* A prompting agent that incorporates self-reflection and iterative improvement over generated outputs (Shinn et al., 2024).
- *PPO:* Proximal Policy Optimization, a classic RL algorithm for policy learning (Schulman et al., 2017).
- *RLOO:* Reinforcement Learning with Offline Observations, a group-based RL approach that estimates advantages without value networks (Kool et al., 2019; Ahmadian et al., 2024).
- *GRPO:* Group-based RL with trajectory-level advantage estimation, designed to scale RL to multi-step tasks (Shao et al., 2024).
- *GiGPO:* Grouped Incremental GPO, a prior hierarchical RL method that performs group-wise advantage estimation for LLM-based agents (Feng et al., 2025b).

### C.2  ENVIRONMENT DETAILS

In each episode, the agent receives a text goal and must accomplish it through multi-turn interaction with the environment. It includes 4,639 task instances across six categories of common household activities: Pick & Place (Pick), Examine in Light (Look), Clean & Place (Clean), Heat & Place (Heat), Cool & Place (Cool), and Pick Two & Place (Pick2). *WebShop* is a complex, web-based interactive environment designed to test the LLM agents in realistic online shopping scenarios. To complete the task, the agent must interact with a simulated HTML-based shopping website to search for, navigate to, and ultimately purchase a suitable item. It contains over 1.1 million products and 12k user instructions, providing a rich and diverse action space.

## C.3 DETAILS OF TRAINING

Notably, we implement GiGPO and HGPO based on the new version of Verl-agent and report the performance in Table 1. Meanwhile, we report them based on the old version of Verl-agent, as shown in Table 7. Generally, we use the same training settings in (Feng et al., 2025b) for fair comparison.

**Hyperparameters for ALFWorld.** All methods are configured with identical hyperparameters: the maximum prompt length is 2048 (4096) tokens, and the maximum response length is 512 tokens. Each episode allows up to 50 environment steps. The learning rate is set to 1e-6 for the actor and 1e-5 for the critic (used only in PPO). We adopt a rule-based reward, assigning a reward of 10 for success and 0 for failure. To handle invalid actions generated by the agent, we apply a reward penalty of -0.1. For all group-based RL methods, we use a group size of 8 and a training size, resulting in a total of $16 \times 8 = 128$ environments. In contrast, PPO uses 128 separate environments for rollouts. The rollout temperature is set to 1.0, while the validation temperature is set to 0.4. The mini-batch size is 256, and the KL-divergence loss coefficient is set to 0.01. The discount factor $\gamma$ is set to 0.95.

---

**Prompt Template for ALFWorld**

You are an expert agent operating in the ALFRED embodied Environment. Your task is to: {task_description}. Prior to this step, you have already taken {step_count} step(s). Below are the most recent {history_length} observations and the corresponding actions you took: {action_history}. You are now at step {current_step} and your current observation is: {current_observation}. Your admissible actions of the current situation are: [{admissible_actions}].

Now it's your turn to take an action. You should first reason step-by-step about the current situation. This reasoning process MUST be enclosed within <think> </think> tags. Once you've finished your reasoning, you should choose an admissible action for current step and present it within <action> </action> tags.

---

Figure 5: The prompt template of ALFWorld agents.

---

**Prompt Template for WebShop**

You are an expert autonomous agent operating in the WebShop e-commerce environment. Your task is to: {task_description}. Prior to this step, you have already taken {step_count} step(s). Below are the most recent {history_length} observations and the corresponding actions you took: {action_history}. You are now at step {current_step} and your current observation is: {current_observation}. Your admissible actions for the current situation are: [{available_actions}].

Now it's your turn to take one action for the current step. You should first reason step-by-step about the current situation, then think carefully which admissible action best advances the shopping goal. This reasoning process MUST be enclosed within <think> </think> tags. Once you've finished your reasoning, you should choose an admissible action for current step and present it within <action> </action> tags.

---

Figure 6: The prompt template used for WebShop agents.

**Hyperparameters for WebShop.** All methods are configured with identical hyperparameters: the maximum prompt length is 4096 tokens, and the maximum response length is 512 tokens. Each episode is limited to 30 environment steps. The learning rate is 1e-6 for the actor and 1e-5 for the critic (used only in PPO). We adopt a rule-based reward, assigning a reward of 10 for success and 0 for failure. Invalid actions are penalized with a reward of -0.1. As with ALFWorld, all group-based RL methods use a group size of 8 and sample 16 groups per rollout, totaling $16 \times 8 = 128$ environments. PPO, on the other hand, uses 128 distinct environments for rollouts. The rollout temperature is set to 1.0, while the validation temperature is set to 0.4. The mini-batch size is 64, and the KL-divergence loss coefficient is set to 0.01. The discount factor $\gamma$ is set to 0.95.

**Computing Details.** Experiments using Qwen2.5-1.5B-Instruct are conducted on two NVIDIA H100 GPUs, while those using Qwen2.5-7B-Instruct are trained on four NVIDIA H100 GPUs. Each experiment is trained for a total of 160 training iterations. The validation data size is 512.

## C.4 TRAINING METRICS

- *Mean Advantages:* This metric shows how much better the chosen actions are compared to the average action. A positive and stable value means the agent usually selects better actions, while large fluctuations suggest unstable training.

- *Policy Gradient Loss:* This loss is the main signal for updating the policy. A smooth and gradually decreasing value indicates stable learning. If the loss becomes too large or changes sharply, it means the updates are too aggressive and may harm training stability.

- *KL Divergence:* KL loss measures how different the new policy is from the old one. It acts as a constraint to prevent the policy from changing too quickly. A moderate KL value means the agent is learning steadily, while a very high value can cause divergence and a very low value may slow down learning.

- *Policy Gradient Clip Fraction:* This metric shows the proportion of gradients that are clipped during optimization. Gradient clipping prevents extreme updates. A moderate fraction suggests stable training, but if the fraction is too high, it means many updates are unstable and are being restricted.

- *Mean Reward:* The mean reward reflects the average return the agent receives per episode. It is a direct measure of progress: higher rewards indicate better performance. If the mean reward increases smoothly, it shows effective learning, while sudden drops suggest instability.

- *Episode Success Rate:* This metric measures the percentage of episodes in which the agent completes the task. It is an intuitive indicator of how well the agent achieves its goal. A rising success rate shows that the agent is improving and that training is effective.

## C.5 PROMPTS

The prompts we use for LLM agents are presented in Figure 5 and Figure 6. These prompt templates are constructed using Python-style string formatting, where placeholders enclosed in curly braces ({}) represent semantic slots. These placeholders, such as {task_description}, {step_count}, and {current_observation}, are dynamically populated at runtime via Python's .format() function. To enrich the agent's context, we use historical information and set the history length to 2.

The <think>...</think> block instructs the agent to perform step-by-step reasoning, thereby promoting chain-of-thought style deliberation explicitly. The <action>...</action> block is used to indicate the final action decision clearly.

## D   MORE EXPERIMENTAL RESULTS

### D.1 THE PERFORMANCE OF USING THE OLD VERSION OF VERL-AGENT

Note that the results reported in the original manuscript (Table 7) were obtained using the earlier version of Verl-agent (paper_version). Following subsequent updates to veRL, we report the performance based on the latest version of Verl-agent in Table 1. Importantly, the updates to Verl-agent (veRL) involve changes to the framework implementation only and do not modify any underlying algorithms. The experimental results remain consistent and continue to demonstrate the superiority of HGPO.

### D.2 TRAINING DYNAMICS

We show training dynamics of HGPO (Red), GiGPO (Yellow), and GRPO (Purple) on WebShop using Qwen2.5-1.5B-Instruct as shown in Figure 8. Figures 7 and 8 illustrate the training dynamics of GRPO, GiGPO, and HGPO across six metrics: mean advantages, policy gradient loss, KL loss, policy gradient clip fraction, mean reward, and episode success rate. Detailed definitions of these metrics are provided in Appendix C.4. Overall, our method achieves more stable and efficient policy optimization. In particular, for the policy gradient clip fraction, HGPO (red curve) maintains a moderate level, suggesting stable training, whereas GiGPO and GRPO display higher fractions, reflecting instability and constraint. For the KL loss, GRPO's curve is too low, indicating slow

Table 7: Performance comparison on ALFWorld and WebShop. Note that we used *the old version of Verl-agent* and updated the performance of using the new version of Verl-agent (with veRL updating) in Table1. For ALFWorld, we report the overall success rate (↑) for both *in-distribution* (In-Success) and *out-of-distribution* tasks (Out-Success). For WebShop (15 steps), we report the average task score (↑) and the average task success rate (↑). Most results are averaged over 3 random seeds during testing. The best results are highlighted in bold.

| Model | Type | Method | ALFWorld | | WebShop | |
| | | | In-Success | Out-Success | Task Scores | Task Success Rates |
|---|---|---|---|---|---|---|
| Q2.5-1.5B | RL Training | GiGPO ($K$=2) | $85.42_{\pm1.32}$ | $80.72_{\pm1.62}$ | $84.52_{\pm0.98}$ | $69.79_{\pm0.59}$ |
| | RL Training | **HGPO** ($K$=2) | $\mathbf{89.58}_{\pm0.45}$ | $\mathbf{80.73}_{\pm2.38}$ | $\mathbf{87.53}_{\pm0.77}$ | $\mathbf{72.66}_{\pm1.78}$ |
| | RL Training | GiGPO ($K$=4) | $85.15_{\pm2.81}$ | $80.98_{\pm0.45}$ | $88.5_{\pm0.49}$ | $74.08_{\pm0.98}$ |
| | RL Training | **HGPO** ($K$=4) | $\mathbf{92.45}_{\pm0.81}$ | $\mathbf{89.06}_{\pm2.34}$ | $\mathbf{88.90}_{\pm0.90}$ | $\mathbf{75.91}_{\pm1.19}$ |
| Q2.5-7B | RL Training | GiGPO ($K$=2) | $89.84_{\pm2.20}$ | $82.81_{\pm5.46}$ | $86.23_{\pm1.43}$ | $75.13_{\pm1.37}$ |
| | RL Training | **HGPO** ($K$=2) | $\mathbf{91.15}_{\pm1.19}$ | $\mathbf{84.89}_{\pm4.30}$ | $\mathbf{88.93}_{\pm0.84}$ | $\mathbf{76.43}_{\pm1.47}$ |
| | RL Training | GiGPO ($K$=4) | $90.88_{\pm0.90}$ | $87.76_{\pm0.45}$ | $87.25_{\pm1.02}$ | $76.18_{\pm1.25}$ |
| | RL Training | **HGPO** ($K$=4) | $\mathbf{94.79}_{\pm0.90}$ | $\mathbf{93.22}_{\pm1.62}$ | $\mathbf{87.88}_{\pm0.41}$ | $\mathbf{77.21}_{\pm0.22}$ |

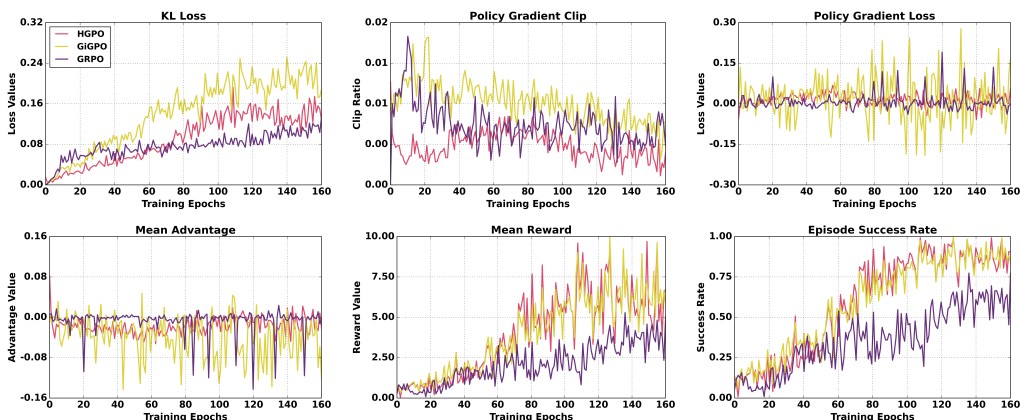

Figure 7: Training dynamics of HGPO (Red), GiGPO (Yellow), and GRPO (Purple) on ALFWorld using Qwen2.5-1.5B-Instruct. The details of these metrics are shown in Appendix D.2.

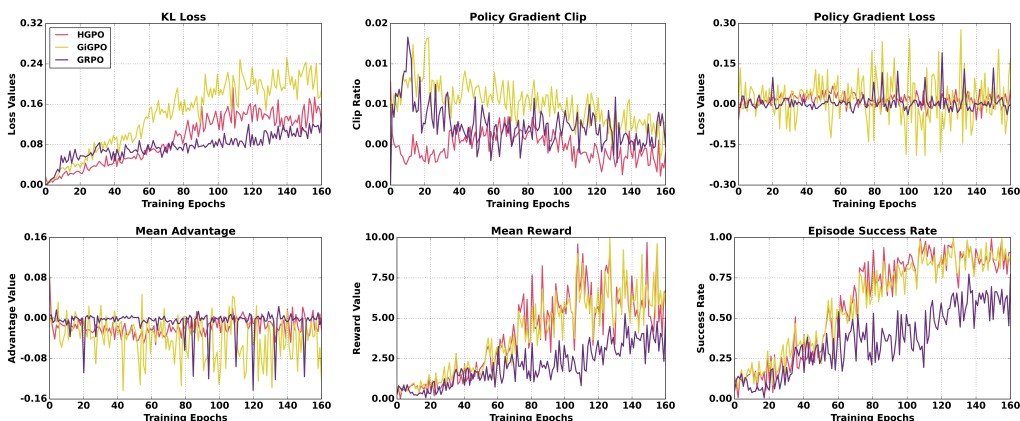

Figure 8: Training dynamics of HGPO (Red), GiGPO (Yellow), and GRPO (Blue) on WebShop using Qwen2.5-1.5B-Instruct. Best viewed in color.

learning, while GiGPO's curve is relatively high, reflecting an overly aggressive learning process. By contrast, HGPO achieves a balanced trajectory, demonstrating steady and stable policy learning. We have also made the training and evaluation Weights & Biases (W&B) logs publicly available.

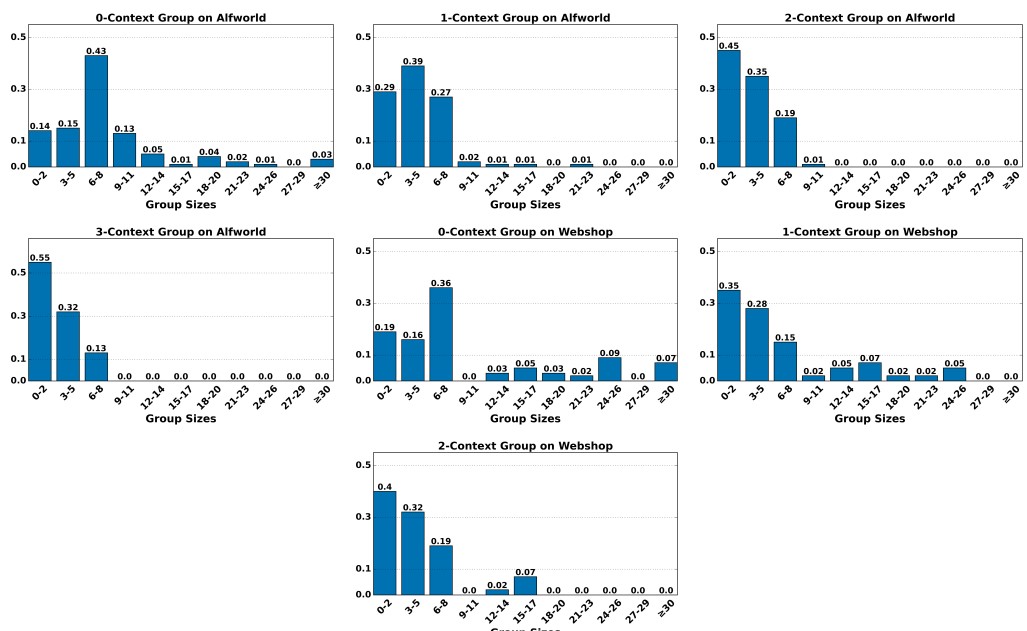

Figure 9: The distributions of hierarchical group sizes ($K = 4$) on ALFWorld and WebShop using Qwen2.5-1.5B-Instruct.

### D.3 THE DISTRIBUTION

We report the distributions of hierarchical group sizes (K = 4) on ALFWorld and WebShop using Qwen2.5-1.5B-Instruct as shown in Table 9.

## E USE OF LLMS

We used LLMs exclusively as writing assistants to refine language. In particular, their use was restricted to grammar correction, style improvement, and phrasing adjustments for clarity and conciseness.

