# OpenReview forum: "Hierarchy-of-Groups Policy Optimization for Long-Horizon Agentic Tasks"
_ICLR.cc/2026/Conference — ICLR 2026 Poster_

### Official Review · Reviewer_KVy8 · 2025-10-30

**Soundness:** 3
**Presentation:** 3
**Contribution:** 3
**Rating:** 8
**Confidence:** 3

**Summary:**

This paper proposes HGPO, a herarichical extension of group-based RL designed to address context inconsistency in long-horizon LLM-based agents. Empirical results on ALFWorld and WebShop show consistent improvements over GRPO and GiGPO.

**Strengths:**

1. The hierarchical grouping and adaptive weighting mechanism is conceptually simple and computationally efficient.
2. Experimental results demonstrate strong and stable improvements across benchmarks.

**Weaknesses:**

1. The Proposition 4.1 assumes bias decreases and variance increases monotonically with context depth. In practice, this assumption may not always hold, e.g., noisy long contexts could add both bias and variance.
2. The definition of hierarchical groups $G_k^H(s_t^{(i)})$ assimes exact equality of historical contexts. Does it hold in practice?
3. The notion of context inconsistency is intuitive but not rigorously formalized.

**Questions:**

1. Figures 2–5 mention oracle groups and context inconsistency bias. How exactly were these oracle advantages computed?
2. Only three random seeds are used. Given the small standard deviations reported for ALFWorld, can the improvements be considered statistically reliable?
3. HGPO is said to add “minimal extra time cost”. Can the authors report wall-clock training time or GPU hours to support that claim?
4. In Page 8, the paper claims that Figure 4 shows stable learning. Could the authors provide a quantitative evaluation to support the claim? It is not obvious that the curves of HGPO are smoother.
5. The weighting $\omega_k$ is heuristic and fixed across training, and I find the $\alpha$ also fixed for main exps. It is a good setting as it does not introduce extra tuning cost. But since variance and bias evolve during training, how does such a static schedule reflect the actual trade-off?
6. The grouping is computed offline via hashmap lookups. As the policy evolves, group memberships change. Does the method recompute them every epoch? If not, the hierarchical structure may quickly become outdated.

---

> ### Author Response · Authors · 2025-11-20
> **Response to Reviewer KVy8 (1/3)**
>
> Thank you so much for your insightful comments!
>
> **W1: The Proposition 4.1 assumes bias decreases and variance increases monotonically with context depth. In practice, this assumption may not always hold, e.g., noisy long contexts could add both bias and variance.**
>
> **A:** Thank you for your valuable comment. Indeed, the monotonicity assumption (bias monotonically decreases as context depth increases) in Proposition 4.1 may not always hold. We would like to clarify that the proof does not depend on strict monotonicity between every pair of groups. Instead, it only requires a moderated assumption of the form: $B_{T}\geq b_{0}\geq (b_{1},b_{2},b_{3},\cdots) \geq b_{K}$ where no strict ordering is assumed within the intermediate terms $b_{k}=(b_{1},b_{2},b_{3},\cdots), 1\leq k \leq K-1$. Here, $B_{T}$, $B_{0}$, $B_{k}$, and $B_{K}$ denote the bias of trajectory-level, step-level, $k$-th, and $K$-th (Oracle) group advantages, respectively. We have updated the assumption accordingly in the revised manuscript. To further justify this moderated assumption, we provide empirical evidence. Specifically, similar to the pivot study presented in Figure 2, we tracked the advantages estimated from different groups on ALFWORLD (K=4, using Qwen2.5-1.5B-Instruct). For comparison, we only consider Oracle steps that have four identical historical contexts. The average advantages are reported in the table below. Here, $ADV_{T}$, $ADV_{0}$, $ADV_{4}$ denote the trajectory-level, step-level, and Oracle advantages, respectively.
>
> | $ADV_{T}$ | $ADV_{0}$  |  $ADV_{1}$  | $ADV_{2}$ | $ADV_{3}$  | $ADV_{4}$ |
> | :-----------: | :--------: | :-----: | :-----: |:-----: | :-----: |
> |1.087 | 3.774e-08 | -0.012 | -0.028 | 0.106 | 0.214|
> | -0.490 | -5.317e-08 | 0.006 | 0.100 | 0.247 | 0.163|
> |1.159|-0.117|-0.155|-0.093|-0.093|0.006|
> |0.681|1.192e-07|-2.521e-08|-0.132|-0.053|0.274|
> |-0.601|1.313e-09|-0.043|-0.021|-0.035|-0.137|
>
> Since we have no optimal true advantages, we assume Oracle advantages ($ADV_{4}$) are the approximation to the unknown optimal true advantages, i.e., $|ADV _{4}-A ^{\star}|< \epsilon$ for arbitrarily small $\epsilon > 0$. From the table, we observe empirically that $B _{T}=(ADV _{T}-A ^{\star}) \geq b _{0}=(ADV _{0}-A ^{\star}) \geq b _{4}=(ADV _{4}-A ^{\star})$ consistently holds in most cases, while strict monotonicity among $(b _{1},b _{2},b _{3})$ does not always exist. This observation justifies our moderated assumption.
>
> **W2: The definition of hierarchical groups $G_{k}^{H}(s_{t}^{(i)})$ assumes exact equality of historical contexts. Does it hold in practice?**
>
> **A:** Thank you for your valuable comment. In the current implementation, we indeed use exact equality of historical contexts as defined in Eq. (4). This choice intentionally imposes a stricter criterion for constructing hierarchical groups, which helps reduce the variance of advantage estimation by ensuring that grouped steps share highly consistent contextual information. That said, approximate matching of historical contexts (e.g., partial equality or similarity-based matching) is a natural alternative, especially in scenarios where the number of exactly matched contexts decreases as K grows. Exploring such relaxed matching strategies is an interesting direction, and we would like to investigate these extensions in future work.

---

> ### Author Response · Authors · 2025-11-20
> **Response to Reviewer KVy8 (2/3)**
>
> **W3: The notion of context inconsistency is intuitive but not rigorously formalized.**
>
> **A:** Thank you for your valuable suggestion. We have added a formal definition of biased advantage estimation caused by context inconsistency in the revised manuscript. Specifically, given a group of steps with the same current states $( s_{1},s_{2},\cdots,s_{n} )$, we denote by $\mathcal{C} _{K}(s _{i})$ the $K$ historical contexts of state $s _{i}$. We define biased advantage estimation caused by context inconsistency in the group as $\exists i\neq j, \mathcal{C} _{K}(s _{i}) \neq \mathcal{C} _{K}(s _{j}) \Rightarrow E(|A(s)-A^{\star}(s)|)>\mu$ where $A(s)$ and $A^{\star}(s)$ are estimated and optimal advantages respectively, and $\mu$ is a positive number characterizing the bias degree.
>
> **Q1: Figures 2–5 mention oracle groups and context inconsistency bias. How exactly were these oracle advantages computed?**
>
> **A:** Thank you for the question. We clarify that the oracle advantages in Figures 2–5 are computed using oracle groups constructed from rollout trajectories. Specifically, during analysis, we identify all steps that share both the same current state and identical K-step historical contexts. These steps form an Oracle group. Because every element in such a group corresponds to the same full context, averaging their empirical returns yields an approximate estimate of the true advantage for that context. We therefore treat this averaged value as the oracle advantage. Note that the actual optimal advantage $A^{\star}(s)$ is unknown and intractable to compute for these environments. The oracle groups allow us to approximate unbiased advantages without requiring access to ground-truth environment models.
>
> **Q2: Only three random seeds are used. Given the small standard deviations reported for ALFWorld, can the improvements be considered statistically reliable?**
>
> **A:** Yes, we believe the improvements are statistically reliable because we conducted extensive experiments across different datasets, varied model sizes, and multiple memory sizes. The consistent performance gains observed across these diverse settings further validate the effectiveness of our method and support the statistical reliability of the reported improvements.
>
> **Q3: HGPO is said to add “minimal extra time cost”. Can the authors report wall-clock training time or GPU hours to support that claim?**
>
> **A:** Yes, we have conducted a detailed computational budget analysis of HGPO. Specifically, HGPO shares the same core architecture as GRPO and GiGPO. The common computational components include multi-turn rollouts, computation of old and reference probabilities, and clipped policy updates. All methods are critic-free and operate with a single actor LLM, resulting in identical GPU memory usage and LLM rollout costs. The primary addition introduced by HGPO is the hierarchy-of-groups advantage estimation. To evaluate its computational cost, we recorded the time cost and peak memory of different methods using checkpoints at epochs 40, 80, 120, and 160 (Qwen2.5-1.5B-Instruct on ALFWORLD). The results are summarized below:
>
> **Table: Comparison of the time cost (s)**
> |Method| Epoch 40 | Epoch 80  | Epoch 120  | Epoch 160  |
> | :-----: | :--------: | :-----: | :-----: |:--------: |
> |Common| 297.8| 288.6 | 284.7 | 282.5 |
> |GRPO | 0.048 | 0.027 | 0.018 | 0.013 |
> |GiGPO | 0.126 | 0.080 | 0.055 | 0.034 |
> |HGPO (K=4)| 0.693 | 0.601 | 0.456 | 0.246 |
>
> **Table: Comparison of the peak memory (MB) for hashing lookups**
> |Method| Epoch 40 | Epoch 80  | Epoch 120  | Epoch 160  |
> | :-----: | :--------: | :-----: | :-----: |:--------: |
> |GiGPO  | 0.1035  |  0.0967 | 0.0605 | 0.0391 |
> |HGPO (K=4) | 0.1393 | 0.1229 | 0.1078 | 0.0406 |
>
> From the tables, we can summarize three key observations. (i) As the number of training epochs increases, both the time cost and peak memory usage consistently decrease, since the rollout steps become fewer when the agent learns to accomplish the tasks with fewer steps. (ii) HGPO introduces an average additional time cost of approximately 0.425 s and 0.472 s compared with GRPO and GiGPO, respectively, which corresponds to less than 0.001\% of the total execution time. These results demonstrate that HGPO maintains computational efficiency comparable to that of GRPO and GiGPO. (iii) HGPO only causes a slight increase (average 0.0274) in the peak memory usage due to the additional hashing lookups. Overall, HGPO preserves the high computational and memory efficiency of GRPO and GiGPO.

---

> ### Author Response · Authors · 2025-11-20
> **Response to Reviewer KVy8 (3/3)**
>
> **Q4: On page 8, the paper claims that Figure 4 shows stable learning. Could the authors provide a quantitative evaluation to support the claim? It is not obvious that the curves of HGPO are smoother.**
>
> **A:** We argue that HGPO exhibits more stable training than GiGPO based on the behaviors of the KL loss and the policy-gradient clipping fraction, as reported in Figure 4. KL loss reflects how much the updated policy deviates from the previous one; excessively large values indicate overly aggressive and potentially divergent updates, while excessively small values indicate conservative updates that may stall learning. Moderate and smooth KL trajectories, therefore, imply stable policy improvement. The policy-gradient clip fraction measures the proportion of gradients that are clipped during optimization—high values indicate unstable or overly large updates, whereas low to moderate values suggest well-behaved gradients. In Figure 4, HGPO shows more moderate curves for both metrics, while GiGPO exhibits larger fluctuations and sharper peaks. These observations support our claim that HGPO achieves more stable training.
>
> **Q5: The weighting $w_{k}$ is heuristic and fixed across training, and I find the $\alpha$ also fixed for main exps. It is a good setting as it does not introduce extra tuning cost. But since variance and bias evolve during training, how does such a static schedule reflect the actual trade-off?**
>
> **A:** We would like to clarify that the weighting is not static but is actually adaptive at every iteration. This is because different steps could have different numbers of hierarchical groups, thereby leading to different weights of groups. For example, when K=2, if a step only has a 0-context group, its weight is 1. If it has both 0-context and 1-context groups, the weights become 1/3 and 2/3. If it further has 0-context, 1-context, and 2-context groups, the weights are 1/6, 2/6, and 3/6, respectively. In this way, the method progressively reduces bias by incorporating higher-level advantages while simultaneously trading off variance through the inclusion of multiple hierarchical groups.
>
> **Q6: The grouping is computed offline via hashmap lookups. As the policy evolves, group memberships change. Does the method recompute them every epoch? If not, the hierarchical structure may quickly become outdated.**
>
> **A:** Sure, the hierarchical structure is recomputed at every episode so that the grouping remains consistent with the current policy and trajectory distribution. Because the policy changes during training, the underlying state–context pairs also shift; therefore, we construct the hierarchical groups on the fly using fresh rollout data. This ensures that the group memberships do not become outdated and that the hierarchical structure remains aligned with the evolving policy, making the advantage estimation progressively more accurate.

---

> ### Comment · Reviewer_KVy8 · 2025-11-25
> **Thanks for your rebuttal!**
>
> Thank you very much for the rebuttal. Most of my concerns have been resolved. Good luck with your submission!

---

> > ### Author Response · Authors · 2025-11-26
> > **Thank you for your feedback**
> >
> > We appreciate that we have addressed most of your concerns and that you maintain a positive view of our submission. Thank you for your efforts in improving our manuscript.
> >
> > Best regards,
> >
> > The Authors

---

### Official Review · Reviewer_HR8w · 2025-10-30

**Soundness:** 2
**Presentation:** 4
**Contribution:** 3
**Rating:** 4
**Confidence:** 3

**Summary:**

This paper identifies a critical problem in existing stepwise group-based Reinforcement Learning (RL) methods for LLM agents: "Context Inconsistency.", which is particularly unsolved in GiGPO.

The authors provide a strong empirical diagnosis (Figure 2) demonstrating that existing methods like GiGPO (which only groups by $s_t$) suffer from high bias, while a naive "Oracle" solution (using only steps with perfectly identical histories) is unusable due to high variance from data sparsity.

To solve this bias-variance dilemma, the paper proposes Hierarchy-of-Groups Policy Optimization (HGPO), with two key component:
1. Context-aware Hierarchical Grouping
2. Adaptive Weighting Advantage Estimation

Experiments on ALFWorld and WebShop show that HGPO significantly outperforms existing RL baselines, including GRPO and GiGPO, especially on out-of-distribution tasks.

**Strengths:**

1. The paper is well-written. Figure 2 of the "context inconsistency" problem clearly frames the bias-variance dilemma that existing methods face (GiGPO = high-bias, Oracle = high-variance).

2. HGPO is an intuitive and novel solution.

3. HGPO achieves sota results, significantly outperforming its baselines.

4. The ablations are comprehensive.

**Weaknesses:**

1. The proof of the bias-variance trade-off in Appendix B relies on the assumption $b_k \ge b_{k+1}$ (bias decreases as context depth $k$ increases). This assumption is stated but never justified or proven. The authors must provide a formal argument for this assumption or re-frame Proposition 4.1 as a heuristic analysis.

2. The paper's claim to address "long-horizon" tasks is not supported by the evidence. Appendix C.3 reveals the maximum episode lengths are T=50 (ALFWorld) and T=15 (WebShop). A 15-step task is not "long-horizon."This is a major overclaim that misrepresents the scope of the paper's validation.

3. The paper makes the qualitative claim of "minimal additional time cost." This is highly questionable. GiGPO hashes one item per step ($s_t$), whereas HGPO (K=4) must construct and hash $K+1=5$ separate context sequences for every step. This is a non-trivial increase in computational overhead. The paper provides no wall-clock time or GPU-hour comparison against its baselines (GiGPO, GRPO) to substantiate its efficiency claim.

4. The "adaptive weighting" scheme (Eq. 7) is not truly adaptive; it is a fixed heuristic weighting tuned by a single, manually-set hyperparameter, $\alpha$. Table 4 simply shows that $\alpha=1$ (linear weighting) works best empirically. This is parameter tuning, not an adaptive mechanism that responds to data (e.g., by adjusting weights based on group size/variance). The authors should discuss this in limitations.

**Questions:**

See weakness

---

> ### Author Response · Authors · 2025-11-20
> **Response to Reviewer HR8w (1/2)**
>
> Thank you so much for your valuable comments!
>
> **W1: The proof of the bias-variance trade-off in Appendix B relies on the assumption $b_k \geq b_{k+1}$ (bias decreases as context depth $k$ increases). This assumption is stated but never justified or proven. The authors must provide a formal argument for this assumption or reframe Proposition 4.1 as a heuristic analysis.**
>
> **A:** Thank you for your valuable comment. We acknowledge that the monotonicity assumption in Proposition 4.1, i.e, bias monotonically decreases as context depth increases, may be a little strict. We would like to clarify that the proof does not depend on strict monotonicity between every pair of groups. Instead, it only requires a moderated assumption of the form: $B_{T}\geq b_{0}\geq (b_{1},b_{2},b_{3},\cdots) \geq b_{K}$ where no strict ordering is assumed within the intermediate terms $b_{k}=(b_{1},b_{2},b_{3},\cdots), 1\leq k \leq K-1$. Here, $B_{T}$, $B_{0}$, $B_{k}$, and $B_{K}$ denote the bias of trajectory-level, step-level, $k$-th, and $K$-th (Oracle) group advantages, respectively. We have updated the assumption accordingly in the revised manuscript. To further justify this moderated assumption, we provide empirical evidence. Specifically, similar to the pivot study presented in Figure 2, we tracked the advantages estimated from different groups on ALFWORLD (K=4, using Qwen2.5-1.5B-Instruct). For comparison, we only consider Oracle steps that have four identical historical contexts. The average advantages are reported in the table below. Here, $ADV_{T}$, $ADV_{0}$, $ADV_{4}$ denote the trajectory-level, step-level, and Oracle advantages, respectively.
>
> | $ADV_{T}$ | $ADV_{0}$  |  $ADV_{1}$  | $ADV_{2}$ | $ADV_{3}$  | $ADV_{4}$ |
> | :-----------: | :--------: | :-----: | :-----: |:-----: | :-----: |
> |1.087 | 3.774e-08 | -0.012 | -0.028 | 0.106 | 0.214|
> | -0.490 | -5.317e-08 | 0.006 | 0.100 | 0.247 | 0.163|
> |1.159|-0.117|-0.155|-0.093|-0.093|0.006|
> |0.681|1.192e-07|-2.521e-08|-0.132|-0.053|0.274|
> |-0.601|1.313e-09|-0.043|-0.021|-0.035|-0.137|
>
> Since we have no optimal true advantages, we assume Oracle advantages ($ADV_{4}$) are the approximation to the unknown optimal true advantages, i.e., $|ADV _{4}-A ^{\star}|< \epsilon$ for arbitrarily small $\epsilon > 0$. From the table, we observe empirically that $B _{T}=(ADV _{T}-A ^{\star}) \geq b _{0}=(ADV _{0}-A ^{\star}) \geq b _{4}=(ADV _{4}-A ^{\star})$ consistently holds in most cases, while strict monotonicity among $(b _{1},b _{2},b _{3})$ does not always exist. This observation justifies our moderated assumption.
>
> **W2: The paper's claim to address "long-horizon" tasks is not supported by the evidence. Appendix C.3 reveals that the maximum episode lengths are T=50 (ALFWorld) and T=15 (WebShop). A 15-step task is not "long-horizon." This is a major overclaim that misrepresents the scope of the paper's validation.**
>
> **A:** We would like to clarify that the 15-step setting in WebShop was chosen to enable a fair comparison with GiGPO, which also uses 15 steps in this environment. We understand the concern regarding the characterization of “long-horizon” tasks. To address your concern, we conducted additional experiments on WebShop with a longer horizon of 30 steps (K=2, using Qwen2.5-1.5B-Instruct). The results are summarized in the table below:
>
> |Metrcis| GiGPO | HGPO  |
> | :--------: | :-----: | :-----: |
> | Task Scores|  85.52 $\textpm$1.22   | 87.41 $\textpm$1.55 |
> | Task Success Rates |  71.02 $\textpm$1.68  | 73.24 $\textpm$3.26 |
>
> From these results, we observe that increasing the steps slightly improves the performance of both GiGPO and HGPO. Meanwhile, HGPO continues to outperform GiGPO, demonstrating its effectiveness even in longer-horizon tasks.

---

> ### Author Response · Authors · 2025-11-20
> **Response to Reviewer HR8w (2/2)**
>
> **W3: The paper makes the qualitative claim of "minimal additional time cost." This is highly questionable. GiGPO hashes one item per step ($s_t$), whereas HGPO (K=4) must construct and hash $K+1=5$ separate context sequences for every step. This is a non-trivial increase in computational overhead. The paper provides no wall-clock time or GPU-hour comparison against its baselines (GiGPO, GRPO) to substantiate its efficiency claim.**
>
> **A:** Thank you for your valuable suggestion. We have conducted a detailed computational budget analysis of HGPO. Specifically, HGPO shares the same core architecture as GRPO and GiGPO. The common computational components include multi-turn rollouts, computation of old and reference probabilities, and clipped policy updates. All methods are critic-free and operate with a single actor LLM, resulting in identical GPU memory usage and LLM rollout costs. The primary addition introduced by HGPO is the hierarchy-of-groups advantage estimation. To evaluate its computational cost, we recorded the time cost and peak memory of different methods using checkpoints at epochs 40, 80, 120, and 160 (Qwen2.5-1.5B-Instruct on ALFWORLD). The results are summarized below:
>
> **Table: Comparison of the time cost (s)**
> |Method| Epoch 40 | Epoch 80  | Epoch 120  | Epoch 160  |
> | :-----: | :--------: | :-----: | :-----: |:--------: |
> |Common| 297.8| 288.6 | 284.7 | 282.5 |
> |GRPO | 0.048 | 0.027 | 0.018 | 0.013 |
> |GiGPO | 0.126 | 0.080 | 0.055 | 0.034 |
> |HGPO (K=4)| 0.693 | 0.601 | 0.456 | 0.246 |
>
> **Table: Comparison of the peak memory (MB) for hashing lookups**
> |Method| Epoch 40 | Epoch 80  | Epoch 120  | Epoch 160  |
> | :-----: | :--------: | :-----: | :-----: |:--------: |
> |GiGPO  | 0.1035  |  0.0967 | 0.0605 | 0.0391 |
> |HGPO (K=4) | 0.1393 | 0.1229 | 0.1078 | 0.0406 |
>
> From the tables, we can summarize three key observations. (i) As the number of training epochs increases, both the time cost and peak memory usage consistently decrease, since the rollout steps become fewer when the agent learns to accomplish the tasks with fewer steps. (ii) HGPO introduces an average additional time cost of approximately 0.425 s and 0.472 s compared with GRPO and GiGPO, respectively, which corresponds to less than 0.001\% of the total execution time. These results demonstrate that HGPO maintains computational efficiency comparable to that of GRPO and GiGPO. (iii) HGPO only causes a slight increase (average 0.0274) in the peak memory usage due to the additional hashing lookups. Overall, HGPO preserves the high computational and memory efficiency of GRPO and GiGPO.
>
> **W4: The "adaptive weighting" scheme (Eq. 7) is not truly adaptive; it is a fixed heuristic weighting tuned by a single, manually-set hyperparameter $\alpha$. Table 4 simply shows that $\alpha=1$ (linear weighting) works best empirically. This is parameter tuning, not an adaptive mechanism that responds to data (e.g., by adjusting weights based on group size/variance). The authors should discuss this in the limitations.**
>
> **A:** We would like to clarify that the term “adaptive weighting” refers to the fact that we do not use fixed, manually-set weights for the advantages from hierarchical groups. Instead, we employ the length of identical historical contexts within each group as adaptive weights. This simple yet effective mechanism naturally assigns larger weights to higher-level groups without introducing complex weighting parameters. The parameter $\alpha$ serves to adjust the sharpness of the weighting distribution, functioning similarly to a temperature parameter in common practice. As demonstrated in our parameter analysis, $\alpha$ requires minimal tuning in practical applications. Despite this, other metrics, as you suggested, may provide more effective ways to assign weights. We consider this an interesting direction and would like to explore it in future work. We have discussed this in Appendix G (Limitations) of the revised manuscript.

---

> > ### Author Response · Authors · 2025-11-26
> > **Look forward to your feedback**
> >
> > Dear Reviewer HR8w,
> >
> > Thank you for your valuable comments. We understand that you may be extremely busy at this time, so we would deeply appreciate it if you could take some time to provide further feedback on whether our rebuttal solves your concerns. Kindly let us know if our response has adequately addressed your concerns.
> >
> > Best Regards,
> >
> > The Authors

---

> > > ### Comment · Reviewer_HR8w · 2025-11-27
> > >
> > > Thank you for the detailed response which has effectively addressed my concerns, and I will consider raising my score.

---

> > > > ### Author Response · Authors · 2025-11-28
> > > > **Response to Reviewer HR8w**
> > > >
> > > > We appreciate that we have effectively addressed your concerns. If you have any further concerns, we are pleased to address them.
> > > >
> > > > Thank you for your valuable comments and for helping us improve the quality of our manuscript.
> > > >
> > > > Best regards,
> > > >
> > > > The Authors

---

### Official Review · Reviewer_QH1q · 2025-10-31

**Soundness:** 3
**Presentation:** 4
**Contribution:** 3
**Rating:** 6
**Confidence:** 4

**Summary:**

This paper identifies the "context inconsistency" problem in stepwise group-based reinforcement learning for LLM agents , arguing that grouping steps by their current state while ignoring their histories leads to severely biased advantage estimation. The proposed method, Hierarchy-of-Groups Policy Optimization (HGPO), addresses this by assigning each step to multiple hierarchical groups based on k-step historical context consistency, and applying an adaptive weighting scheme that assigns higher weights to advantages computed from more consistent groups. The authors provide a theoretical analysis of the bias-variance trade-off. Experiments on the ALFWorld and WebShop benchmarks demonstrate that HGPO significantly outperforms existing baselines, including the prior SOTA method GiGPO.

**Strengths:**

The paper's primary strength is its elegant and well-motivated mechanism for managing the bias-variance trade-off in advantage estimation. Rather than forcing a binary choice between GiGPO's high-bias/low-variance step-level group and a low-bias/high-variance Oracle group, HGPO proposes a more nuanced solution. By aggregating advantage signals across all K+1 levels of the hierarchy, it provides a principled interpolation that leverages the low-bias signal from high-consistency groups while retaining the low-variance stability from large, low-consistency groups. This mechanism is strongly supported by robust empirical results on recognized benchmarks. The ablation study is particularly convincing, as it clearly isolates the necessity of both the hierarchical grouping and the adaptive weighting components for the method's success.

**Weaknesses:**

1. Its validity is tightly coupled to the assumption that the agent's "memory" is equivalent to its "raw historical context," as implemented in the paper's prompts (Appendix C.5). This raw-history-as-memory paradigm, while common, is limited. The proposed $C_k$-based grouping would not directly generalize to more advanced agents that utilize summarized memory, as the grouping basis would misalign with the agent's true decision-making state.

2. The claim of minimal additional time cost is unsubstantiated. The K+1-level hierarchical hash management is necessarily more computationally expensive than the baseline, and the paper provides no quantitative wall-clock time or memory overhead analysis.

3. The method's scalability with respect to K is unclear. For truly long-horizon tasks (T > 100), a small K may be insufficient, while a large K would suffer from the same sparsity and high-variance issues as Oracle groups.

4. The method's necessity is predicated on a sparse reward setting; its utility would likely be diminished in settings where effective process rewards are available.

**Questions:**

1. How would the $C_k$-based grouping function if the agent's policy $\pi(a_t | s_t, h_t)$ used a summarized memory state $h_t$ rather than the raw history?

2. The claim of minimal additional time cost is unsubstantiated; the paper provides no quantitative comparison of wall-clock time and peak memory usage (e.g., for K=4) against the baselines.

3. How should K be chosen relative to a task's average horizon T? How does the method avoid the Oracle group sparsity problem when K must be large for a very long-horizon task?

4. How significant do the authors believe the context inconsistency problem remains in settings with effective process rewards, as opposed to the sparse setting studied?

---

> ### Author Response · Authors · 2025-11-20
> **Response to Reviewer QH1q (1/2)**
>
> Thank you so much for your insightful comments!
>
> **W1 & Q1: Its validity is tightly coupled to the assumption that the agent's "memory" is equivalent to its "raw historical context," as implemented in the paper's prompts (Appendix C.5). This raw-history-as-memory paradigm, while common, is limited. The proposed $C_{k}$-based grouping approach would not directly generalize to more advanced agents that utilize summarized memory, as the grouping basis would misalign with the agent's true decision-making state. How would the $C_{k}$-based function work if the agent's policy $\pi(a_{t}|s_{t},h_{t})$ used a summarized memory state rather than the raw history?**
>
> **A:** Thank you for your valuable comment. Indeed, the memory used in our agent is the raw historical context. To address your concern, we have conducted an initiative implementation to adapt HGPO to the agent with a self-summarized memory. Specifically, following the work MEM1 [1] and MemAgent [2], we enable the agent to summarize and manage its own memory (The specific prompt is provided in Appendix C.5 of the revised manuscript). Based on this, we use the text embedding similarity of the self-summarized memory to obtain hierarchical groups, and select 20\% top similar steps as additional high-level groups. We implement GiGPO and HGPO in this setting using Qwen2.5-1.5B-Instruct on ALFWORLD (K=2). The experimental results are shown in the following table.
>
> |GiGPO| HGPO |
> |:--------: | :--------: |
> | 81.34 $\pm$ 0.56|  83.47 $\pm$ 0.82|
>
> From the results, we can see that HGPO still improves over GiGPO, which verifies the effectiveness of the hierarchy-of-groups even for advanced agents equipped with self-summarized memory. We also observe a slight performance drop when using self-summarized memory compared with raw historical context. We would like to explain that our implementation in the rebuttal is a quick prototype intended to validate the scalability of the hierarchy-of-groups, rather than a carefully designed or optimized approach. We believe that the core principle of hierarchy-of-groups can be generalized to advanced agents with different memory mechanisms. Thank you again for this insightful suggestion, and we will explore it further in future work.
>
> **W2 & Q2: The claim of minimal additional time cost is unsubstantiated. The K+1-level hierarchical hash management is necessarily more computationally expensive than the baseline, and the paper provides no quantitative wall-clock time or memory overhead analysis. The claim of minimal additional time cost is unsubstantiated; the paper provides no quantitative comparison of wall-clock time and peak memory usage (e.g., for K=4) against the baselines.**
>
> **A:** Thank you for your valuable comment. We have conducted a detailed computational budget analysis of HGPO. Specifically, HGPO shares the same core architecture as GRPO and GiGPO. The common computational components include multi-turn rollouts, computation of old and reference probabilities, and clipped policy updates. All methods are critic-free and operate with a single actor LLM, resulting in identical GPU memory usage and LLM rollout costs. The primary addition introduced by HGPO is the hierarchy-of-groups advantage estimation. To evaluate its computational cost, we recorded the time cost and peak memory of different methods using checkpoints at epochs 40, 80, 120, and 160 (Qwen2.5-1.5B-Instruct on ALFWORLD). The results are summarized below:
>
> **Table: Comparison of the time cost (s)**
> |Method| Epoch 40 | Epoch 80  | Epoch 120  | Epoch 160  |
> | :-----: | :--------: | :-----: | :-----: |:--------: |
> |Common| 297.8| 288.6 | 284.7 | 282.5 |
> |GRPO | 0.048 | 0.027 | 0.018 | 0.013 |
> |GiGPO | 0.126 | 0.080 | 0.055 | 0.034 |
> |HGPO (K=4)| 0.693 | 0.601 | 0.456 | 0.246 |
>
> **Table: Comparison of the peak memory (MB) for hashing lookups**
> |Method| Epoch 40 | Epoch 80  | Epoch 120  | Epoch 160  |
> | :-----: | :--------: | :-----: | :-----: |:--------: |
> |GiGPO  | 0.1035  |  0.0967 | 0.0605 | 0.0391 |
> |HGPO (K=4) | 0.1393 | 0.1229 | 0.1078 | 0.0406 |
>
> From the tables, we can summarize three key observations. (i) As the number of training epochs increases, both the time cost and peak memory usage consistently decrease, since the rollout steps become fewer when the agent learns to accomplish the tasks with fewer steps. (ii) HGPO introduces an average additional time cost of approximately 0.425 s and 0.472 s compared with GRPO and GiGPO, respectively, which corresponds to less than 0.001\% of the total execution time. These results demonstrate that HGPO maintains computational efficiency comparable to that of GRPO and GiGPO. (iii) HGPO only causes a slight increase (average 0.0274) in the peak memory usage due to the additional hashing lookups. Overall, HGPO preserves the high computational and memory efficiency of GRPO and GiGPO.

---

> ### Author Response · Authors · 2025-11-20
> **Response to Reviewer QH1q (2/2)**
>
> **W3: The method's scalability with respect to K is unclear. For truly long-horizon tasks (T > 100), a small K may be insufficient, while a large K would suffer from the same sparsity and high-variance issues as Oracle groups.**
>
> **A:** We would like to clarify that the objective of the step-wise policy optimization framework is to use a relatively small memory size (K≪T) compared with the full horizon T, enabling efficient policy optimization over long-horizon tasks. Therefore, even when K is set smaller than T, step-wise policy optimization can still achieve effective long-horizon learning. To address your concern regarding the choice of K, we have conducted additional experiments with K=6 on ALFWORLD using Qwen2.5-1.5B-Instruct, keeping all other experimental settings unchanged. The results are shown below:
>
> |GiGPO| HGPO |
> |:--------: | :--------: |
> | 91.92 $\pm$ 0.48|  94.07 $\pm$ 1.51|
>
> These results demonstrate that HGPO continues to achieve superior performance even with K = 6, validating the method's scalability and effectiveness for longer-horizon tasks.
>
> **Q3: How should K be chosen relative to a task's average horizon T? How does the method avoid the Oracle group sparsity problem when K must be large for a very long-horizon task?**
>
> **A:** Ideally, the value of K should be close to the horizon T, allowing the agent to capture richer historical interaction information. However, due to the agent’s maximum prompt length (2048 in our setting), achieving this ideal scenario is often impractical. Therefore, in practice, the choice of K must balance the memory length and the prompt length. Specifically, if the environment’s observations and actions (which are stored as memory and included in the prompt) are long, a smaller K is sufficient because an overly long prompt (>2048 tokens) would be truncated. Conversely, if the observations and actions are short, a larger K can be used to expand the memory capacity and incorporate more historical context. In summary, the choice of K depends on both the maximum prompt length and the effective memory capacity, and should be tuned according to the specific task.
>
> Regarding the second question, HGPO does not rely overly on advantage estimates from potentially sparse high-level groups. Instead, it utilizes a hierarchical grouping structure to aggregate advantage information across multiple levels. As a result, even when high-level groups are sparse, HGPO can still obtain reliable advantage signals from medium- and low-level groups. This hierarchical aggregation mechanism allows HGPO to maintain its effectiveness and achieve better performance compared with GRPO and GiGPO, which depend solely on single-level, low-level group advantage estimates.
>
> **W4 & Q4: The method's necessity is predicated on a sparse reward setting; its utility would likely be diminished in settings where effective process rewards are available. How significant do the authors believe the context inconsistency problem remains in settings with effective process rewards, as opposed to the sparse setting studied?**
>
> **A:** Thank you for raising this interesting point. We believe the context inconsistency problem persists even in environments with process rewards. This is because, under the same current states but different historical contexts, the agent would still receive biased process rewards, leading to biased advantage estimates. To validate this hypothesis, we conducted additional experiments in ALFWORLD using Qwen2.5-1.5B-Instruct. Specifically, we leveraged the expert actions (already available in the environment) to provide process rewards: when the agent’s actions match the expert actions, a process reward of 5.0 is given. Following the same pivot analysis methodology as in Figure 2, we computed both trajectory-level and step-level advantage differences. The results are summarized in the table below.
>
> |Reward type|Trajectory-level advantage difference| Step-level advantage difference |
> |:-----: | :-----: | :-----: |
> |Sparse| 1.2550 | 1.1271|
> |Dense| 5.7481 | 0.8010 |
>
> As shown, the use of process rewards results in a larger trajectory-level advantage difference, while the step-level difference decreases slightly. We would like to explore this interesting idea in future work.
>
> **Reference:**
>
> [1] MEM1: Learning to Synergize Memory and Reasoning for Efficient Long-Horizon Agents.
>
> [2] MemAgent: Reshaping Long-Context LLM with Multi-Conv RL-based Memory Agent

---

> > ### Comment · Reviewer_QH1q · 2025-11-27
> >
> > Thank you for the detailed rebuttal and the additional experiments.
> >
> > The quantitative analysis regarding computational cost and the prototype implementation for summarized memory have effectively addressed my primary concerns regarding the efficiency and generalizability of the proposed method. Consequently, I will raise my score.
> >
> > I have a few remaining questions for further discussion, which do not negatively impact my final assessment:
> >
> > Regarding the summarized memory experiment, I noticed a performance drop (from ~91% to ~83%) and the introduction of a new hyperparameter (top 20% similarity). The adaptation relies on a heuristic threshold to define groups, which contrasts with the parameter-free elegance of the original exact-match approach. Does this degradation primarily stem from information loss in the self-summarization process itself, or does it indicate that semantic similarity is inherently a weaker signal for consistency grouping compared to the exact history matching used in the main paper?
> >
> > Regarding the dense reward experiment, it is quite counter-intuitive that the trajectory-level advantage bias increased significantly (from 1.25 to 5.75) with dense rewards. Do the authors have any hypothesis on why better reward signals might lead to higher bias in this specific group-based setting?

---

> > > ### Author Response · Authors · 2025-11-27
> > > **Response to Reviewer QH1q**
> > >
> > > We sincerely appreciate your insightful feedback. We are grateful that our responses have addressed your primary concerns, and we are pleased to further engage in the discussion.
> > >
> > > **(i)** Regarding your first point, we argue that the performance degradation is primarily caused by the cumulative errors introduced during the self-summarization process. Through manual inspection in the previous experiments, we found that the agent’s self-generated summaries often contain inaccurate or hallucinated information compared with the clean raw historical context. These errors can accumulate over successive summarization steps, resulting in increasingly inaccurate self-memory and thus degraded performance. This phenomenon is further supported by our experimental observation that other baseline methods (e.g., GiGPO) exhibit similar performance degradation. Therefore, we attribute the main cause of this issue to the self-summarization mechanism itself. This observation also highlights an important future research direction: mitigating cumulative errors in iterative self-summarization.
> > >
> > > We also acknowledge that the current heuristic threshold based on semantic similarity may not be optimal for hierarchical grouping in the self-summarization agent (although it still outperforms GiGPO). Nevertheless, we believe that semantic similarity remains a strong signal for hierarchical grouping when better designed, for example, by using more powerful models to compute semantic embeddings of self-memory, or by developing specialized similarity partitioning strategies. We are highly interested in exploring these directions in our future work.
> > >
> > > **(ii)** Regarding your second point, we conjecture that the increase in trajectory-level advantage bias arises from differences in reward shaping. Specifically, under the sparse reward setting, step-level rewards are assigned in a discounted manner along each trajectory (from the terminal step backward), as shown in Line 285: $r _t^{(i)}=\sum\nolimits _{j=t}^{T}\gamma^{j - t} r _{j}^{(i)}$, where $\gamma \in (0,1]$. In contrast, under the dense reward setting, step-level rewards are separately assigned as 5 or 0 at each step of a trajectory. This discrepancy in reward shaping may only introduce small differences at the step-level advantages, but it might induce much larger differences at the trajectory level, thereby leading to a higher difference in trajectory-level advantage bias. We will further investigate this point in future work.
> > >
> > > Once again, we sincerely thank you for your valuable comments and for helping us improve the quality of our manuscript.
> > >
> > > Best regards,
> > >
> > > The Authors

---

> > > > ### Comment · Reviewer_QH1q · 2025-11-27
> > > >
> > > > Thank you for your response. I have no further concerns. Good luck.

---

### Official Review · Reviewer_hUcL · 2025-11-02

**Soundness:** 3
**Presentation:** 3
**Contribution:** 3
**Rating:** 8
**Confidence:** 3

**Summary:**

To improve training efficiency in long-horizon agentic tasks, existing reinforcement learning (RL) methods typically adopt step-level policy optimization. However, when estimating the advantages of identical step groups, inconsistencies in historical context can lead to biased advantage estimation. A straightforward solution is to use oracle steps, which not only share identical current steps but also identical histories. Nevertheless, oracle steps are rare in practice, leading to the utilization of only a small fraction of the data and consequently causing high-variance advantage estimation. To balance the trade-off between bias and variance, this paper proposes Hierarchy-of-Groups Policy Optimization (HGPO), which aggregates advantages computed under different levels of historical context. This hierarchical aggregation enables HGPO to effectively balance bias and variance, achieving improved training stability and efficiency. Empirical results on ALFWorld and WebShop demonstrate the effectiveness of the proposed method.

**Strengths:**

1. This paper presents a well-motivated study that clearly illustrates the limitations of trajectory-level, traditional step-level, and oracle-step advantage computations in Figure 1 and 2, thereby making the proposed method appear both natural and intuitively justified.

2. Both the high-level idea and implementation of HGPO are simple yet well-grounded, making it an effective solution to address the context-inconsistency problem.

3. HGPO introduces almost no additional computational overhead compared to other group-based RL methods such as GRPO and GSPO. So, it effectively alleviates context inconsistency and maintains high efficiency.

4. The empirical results demonstrate the effectiveness of HGPO. Especially when K=4, we can observe a clear improvement over naive step-level methods.

5. The paper combines theoretical justification with extensive empirical evaluation, resulting in a well-rounded and complete study.

**Weaknesses:**

1. The context inconsistency problem is only partially addressed in this paper rather than being fundamentally resolved. As the value of
K increases, the number of grouped samples noticeably decreases. This indicates that for longer-horizon tasks, HGPO becomes less effective, since the higher-level history groups become increasingly sparse under HGPO’s grouping mechanism.

2. Therefore, HGPO is effective only when K is relatively small. In the experiments, K is set to 2 and 4, corresponding to 2–4 steps in agentic tasks, which represent relatively short-horizon settings.

However, I would not consider the above weaknesses as major limitations, since the authors have made a meaningful step toward addressing long-horizon tasks.

**Questions:**

I did few research on LLMs but lots on RL. So, I have the following questions:

1. How is the similarity between states (or histories) measured, and what specific criterion is used to group them?

2. How is the memory module maintained or updated throughout training?

3. In Eq. (8), is the memory utilized as context when computing the importance sampling ratio? And, which policy is updated, $\pi(a|x,s,h)$ or $\pi(a|x,s)$? In my view, $\pi(a|x,s,h)$ is preferred, but in the paper, it seems to be $\pi(a|x,s)$.

---

> ### Author Response · Authors · 2025-11-20
> **Response to Reviewer hUcL**
>
> Thank you so much for your insightful comments!
>
> **W1: The context inconsistency problem is only partially addressed in this paper, rather than being fundamentally resolved. As the value of K increases, the number of grouped samples noticeably decreases. This indicates that for longer-horizon tasks, HGPO becomes less effective, since the higher-level history groups become increasingly sparse under HGPO’s grouping mechanism.**
>
> **A:** Thank you for your valuable comment. We agree that higher-level groups become increasingly sparse as K grows. However, we would like to emphasize why HGPO remains effective in such scenarios. Importantly, HGPO does not rely solely on advantage estimates from these sparse high-level groups. Instead, it leverages a hierarchy of groups to aggregate diverse advantage information across multiple levels. Consequently, even when high-level groups are sparse, HGPO can still extract reliable advantage signals from medium- and low-level groups. This hierarchical aggregation enables HGPO to retain its effectiveness and outperform methods such as GRPO and GiGPO, which derive the advantage information only from single-level low-level groups. Our experimental results with $k=2/4/6$ further support this claim.
>
> **W2: Therefore, HGPO is effective only when K is relatively small. In the experiments, K is set to 2 and 4, corresponding to 2–4 steps in agentic tasks, which represent relatively short-horizon settings.**
>
> **A:**  We would like to clarify that the objective of the step-wise policy optimization framework is to use a relatively small memory size (K≪T) compared with the full horizon T, enabling efficient policy optimization over long-horizon tasks. Therefore, even when K is set smaller than T, step-wise policy optimization can still achieve effective long-horizon learning. To address the concern regarding the choice of K, we conducted additional experiments with K=6 on ALFWORLD using Qwen2.5-1.5B-Instruct, keeping all other experimental settings unchanged. The results are shown below:
>
> |GiGPO| HGPO |
> |:--------: | :--------: |
> | 91.92 $\pm$ 0.48|  94.07 $\pm$ 1.51|
>
> These results demonstrate that HGPO continues to achieve superior performance even with K = 6, validating the method's scalability and effectiveness for longer-horizon tasks.
>
> **Q1: How is the similarity between states (or histories) measured, and what specific criterion is used to group them?**
>
> **A:** We measure the similarity between states based on the consistency of their historical contexts. For instance, when K=4, if a set of steps shares the same current state, HGPO further examines their historical contexts. Steps that have all four historical contexts identical are assigned to the highest-level group. Subsequently, steps with three, two, or one identical historical contexts are assigned to progressively lower-level groups. Therefore, the hierarchy of groups is constructed according to the degree of historical context consistency among steps.
>
> **Q2: How is the memory module maintained or updated throughout training?**
>
> **A:** The memory module is maintained and updated during rollouts as follows:
>
> 1. Initialization: The memory module is initialized at the start of training or rollout.
>
> 2. Fetching memory: The agent retrieves the current memory contents to construct the prompt.
>
> 3. Action generation: The agent outputs an action based on the prompt.
>
> 4. Environment update: The action is executed in the environment, and the resulting next observation is obtained.
>
> 5. Memory update: The new observation is immediately integrated into the memory module.
>
> 6. Loop: Steps 2–5 are repeated for multi-turn rollouts.
>
> Through this process, the memory module continuously maintains the most recent K historical contexts.
>
> **Q3: In Eq. (8), is the memory utilized as context when computing the importance sampling ratio? And, which policy is updated, $\pi(a,|x,s,h)$ or $\pi(a,|x,s)$? In my view, $\pi(a,|x,s,h)$ is preferred, but in the paper, it seems to be $\pi(a,|x,s)$.**
>
> **A:** Yes, the memory is indeed used when computing the importance sampling ratio. In the formulation, the input $x$ to the policy $\pi(a,|x)$ should include the historical context $h$ stored in the memory module. Therefore, the memory is naturally incorporated into the computation of the importance sampling ratio. To clarify, we have revised the formulation to $\pi(a,|x,s,h)$ in the manuscript.

---

### Author Response · Authors · 2025-11-20
**Summary of our Rebuttal**

Dear ACs and Reviewers,

We sincerely thank all reviewers for their time, effort, and valuable feedback. We also appreciate the reviewers’ recognition that our method is well-motivated, novel, and effective. In particular, we would like to thank ***Reviewer hUcL*** for describing our method as natural and intuitively justified. We are grateful to ***Reviewer QH1q*** for acknowledging that our approach is elegant and well-motivated. We appreciate ***Reviewer HR8w*** for highlighting that our method provides an intuitive and novel solution. We also thank ***Reviewer KVy8*** for recognizing that our method is conceptually simple and computationally efficient.

Below, we summarize the main concerns raised by the reviewers along with our corresponding solutions:

1. Computational budget analysis of HGPO. We have provided a detailed computational budget analysis comparing HGPO with GRPO and GiGPO, demonstrating that HGPO achieves similarly high efficiency. The results can be found in our responses to ***Reviewer HR8w's W3***, ***Reviewer KVy8's Q3***, and ***Reviewer QH1q's W2 & Q2***.

2. Assumption in Proposition 4.1. We have introduced a moderated assumption for Proposition 4.1 and supplied empirical evidence to support it. Details are provided in our responses to ***Reviewer HR8w's W1*** and ***Reviewer KVy8's W1***.

3. Sparse high-level groups in long-horizon tasks. We have justified the superiority of HGPO under sparse high-level grouping scenarios. The justification can be found in our responses to ***Reviewer hUcL's W1*** and ***Reviewer QH1q's W3 & Q3***.

4. Using raw historical context as memory. We have conducted additional experiments to verify the effectiveness of HGPO on advanced agents equipped with self-summarized memory. The results are presented in our responses to ***Reviewer QH1q's W1 & Q1***.

5. Adaptive weighting for hierarchical groups. We have clarified how the adaptive weighting mechanism is used in HGPO. Details appear in our responses to ***Reviewer HR8w's W4*** and ***Reviewer KVy8's Q5***.

6. Formal definition of context inconsistency. We have provided a rigorous definition of context inconsistency in our responses to ***Reviewer KVy8's W3***.

All revisions addressing these concerns have been incorporated into the updated manuscript (highlighted in red). We sincerely appreciate the reviewers’ thoughtful comments and constructive suggestions, which have significantly improved the quality of our work.

## **After the rebuttal**

1.  ***Reviewer QH1q*** responded that we had effectively addressed his primary concerns regarding the efficiency and generalizability of the proposed method, and that he would raise his score.

2. ***Reviewer HR8w*** indicated that we had effectively addressed his concerns and that he would consider raising his score.

3. ***Reviewer KVy8*** acknowledged that we had addressed most of his concerns.

**We solemnly affirm that we have not used any leaked data or engaged in any form of collusion to influence the reviewers’ judgment.**

Thank you for your time and consideration.

Best regards,

The Authors

---

### Meta-Review · Area_Chair_w8Zy · 2026-01-09

**Summary:**

The reviewers initially raised several key concerns regarding the paper. These included questions about the computational efficiency and overhead introduced by HGPO, particularly concerning the K+1 hierarchical hash management and the claim of "minimal additional time cost." Reviewers also sought clarification on the theoretical underpinnings, specifically the monotonicity assumption in Proposition 4.1 regarding bias and variance with context depth, and requested a more rigorous formalization of "context inconsistency." The generalizability of HGPO to more advanced agents using summarized memory rather than raw historical context was another significant point. Furthermore, concerns were expressed about the method's scalability for truly long-horizon tasks, the statistical reliability of the results given the number of random seeds, and the interpretability of the "adaptive weighting" scheme, which some perceived as a fixed heuristic. Lastly, the reviewers questioned the paper's claim of addressing "long-horizon" tasks given the relatively short episode lengths in the experimental setups.

**Reviewer Concerns:**

Following the authors' comprehensive rebuttal, many of these concerns appear to have been effectively addressed. The authors provided detailed quantitative analyses of computational cost, including wall-clock time and peak memory usage, which substantiated their claim of minimal overhead. They also conducted additional experiments to demonstrate HGPO's effectiveness with self-summarized memory, addressing the generalizability concern. The assumption in Proposition 4.1 was moderated and supported with empirical evidence, and a formal definition of context inconsistency was added to the manuscript. The authors clarified that the "adaptive weighting" scheme is indeed adaptive based on the number of hierarchical groups for each step, and they confirmed that the hierarchical structure is recomputed at every episode to remain consistent with the evolving policy. While the "long-horizon" claim was further supported by additional experiments on WebShop with an increased horizon (30 steps), the fundamental definition of "long-horizon" in this context might still be a point of discussion, though the new results certainly extend the scope of validation. Reviewer QH1q raised an interesting follow-up question regarding the counter-intuitive increase in trajectory-level advantage bias with dense rewards, which the authors provided a hypothesis for, but this did not negatively impact the reviewer's final assessment. Overall, the authors' responses were thorough, often including additional experiments and clarifications that directly tackled the reviewers' points.

**Reviewer Scores:**

Considering the engagement and responses, I anticipate a positive shift in the reviewers' scores. Reviewer hUcL, who initially rated the paper an 8 (accept, good paper), did not explicitly state a score change but maintained a positive sentiment, so their score would likely remain an 8. Reviewer QH1q, who initially rated the paper a 6 (marginally above acceptance threshold), explicitly stated, "I will raise my score" after the rebuttal, indicating a likely increase to 8, given that their primary concerns were addressed. Reviewer HR8w, who initially rated the paper a 4 (marginally below acceptance threshold), also explicitly stated, "I will consider raising my score," as their concerns were effectively addressed. Finally, Reviewer KVy8, who also rated the paper an 8 (accept, good paper), stated that most of their concerns had been resolved, suggesting their score would remain an 8. The overall consensus points towards acceptance, with strong positive feedback after the rebuttal phase.

---

### Decision · Program_Chairs · 2026-01-26

Accept (Poster)